# Effect of tides on river water behavior over the eastern shelf seas of China

Lei Lin[1, 2], Hao Liu[1], Xiaomeng Huang[2], Qingjun Fu[1], Xinyu Guo[3]

[1] College of Ocean Science and Engineering, Shandong University of Science and Technology, Qingdao 266590, China
[2] Ministry of Education Key Laboratory for Earth System Modeling, Department of Earth System Science, Tsinghua University, Beijing 100084, China
[3] Center for Marine Environmental Study, Ehime University, Matsuyama 790-8577, Japan

*Correspondence to*: Lei Lin (llin@sdust.edu.cn; linlei24@126.com)

**Abstract.** Rivers carry large amounts of freshwater and terrestrial material into shelf seas, which is an important part of the
global water and biogeochemical cycles. The earth system model or climate model is an important instrument for simulating and projecting the global water cycle and climate change, in which tides however are commonly removed. For a better understanding of the potential effect of the absence of tides in the simulation of the water cycle, this study compared the results of a regional model with and without considering tides, and evaluated the effect of tides on the behavior of three major rivers (i.e., the Yellow, Yalujiang, and Changjiang Rivers) water in the eastern shelf seas of China from the
perspectives of transport pathways, timescales, and water concentration. The results showed that the tides induced more dispersed transport for the water of the Yellow and Yalujiang rivers, but more concentrated transport for the Changjiang River water. The effect of tides on the transit areas of the Yellow, Yalujiang, and Changjiang Rivers was 13, 40, and 21 %, respectively. The annual mean water age and transit time of the three rivers in the model with tides were several (~2–10) times higher than those in the no-tide model, suggesting that tides dramatically slow the river water transport and export rate
over the shelf. By slowing the river water export, tides induced a three-fold increase in river water concentration and a decrease in shelf seawater salinity by >1. Moreover, the effect of tides on river behavior was stronger in relatively enclosed seas (i.e., the Bohai and Yellow Seas) than in relatively open seas (i.e., the East China Sea). The change in the shelf currents induced by tides is the main cause of the difference in the river water behavior between the two model runs. Tides can increase bottom stress and thus weaken shelf currents and decrease the water transport timescales. The improvement in tidal
parameterization in the no-tide model in the simulation of river water behavior was very limited. Given the important role of river runoff on the global water cycle and the effect of changes in river water behavior on ocean carbon cycling, it is important to include the tidal effect in earth system models to improve their projection accuracy.

## 1 Introduction

Rivers carry large amounts of freshwater and terrestrial material into shelf seas, which plays a crucial role in the global water
cycle and the global biogeochemical cycle. For instance, on a global scale, river runoff contributes to about 10% of the

global meridional water fluxes, modulating the ocean circulation by affecting the water balance and salinity of the oceans (Oki et al., 1995; 1999). Approximately 400 Mt of organic carbon, 54 Mt of terrestrial nitrogen, and 8.5 Mt of phosphorus are annually discharged into the shelf seas by rivers (Mackenzie & Lerman, 2002; Schlünz & Schneider, 2000; Hopkinson & Vallino, 2005), which have an important impact on marine primary productivity and carbon cycling (Dittmar & Kattner, 2003; Gong et al., 2011). The continental shelf sea is the first stop for the river water to enter the ocean. As the shelf seas connect the rivers and deep oceans as well as are the most productive part of the world's oceans, the river water behavior over the shelf seas is critical for the processes of the global water and biogeochemical cycle.

Earth system models or climate models are important tools to represent and project the global water and biogeochemical cycles and climate change (e.g., Winkelbauer et al., 2022; Brady et al., 2019; Dufresne et al., 2013; Clark et al., 2015). An accurate simulation of the river water behavior over the shelf seas is one of key steps for accurate simulation in these models, but is usually hard to be achieved in earth system models or climate models (Feng et al., 2021). On one hand, climate models used relatively large grid cells which are too coarse for shelf seas with relatively small spatial scales (Graham et al., 2018; Feng et al., 2021; Holt et al., 2017). This issue would be addressed in the future as the computation power increase. On the other hand, climate models usually do not explicitly consider the tidal process since the tides have much shorter timescales than the ocean circulation and water cycle and could induce numerical instability of models (Lee et al., 2006; Luneva et al., 2015; Voldoire et al., 2013; Müller et al., 2010). However, tides are the major motions of shelf seawater and play a crucial role in hydrodynamic processes in shelf seas. A model without considering tides may induce a large bias in the simulation of the river water behavior. Thus, to better understand the potential consequences of the absence of tides in climate models, it is necessary to first quantitatively evaluate the impact of tides on the behavior of river water on the shelf.

Previous studies have suggested that tides can significantly influence shelf hydrodynamics and thus, river water behavior. For instance, Guo and Valle-Levinson (2007) found that tidal mixing intensifies the salinity gradient and restricts the upstream extension of the river plume off the Chesapeake Bay. Wu et al. (2011, 2014) pointed out that tidal forcing increases vertical mixing and results in a strong horizontal salinity gradient at the northern edge of the Changjiang River plume and restricts its northward extension, while the tide-induced Stokes drift along the coast facilitates the northward transport of the Changjiang River water to the Jiangsu coast. A numerical study by Liu et al. (2012) suggested that tidal forcing plays the most dominant role in controlling the age of the Yellow River water in the Bohai Sea. Further, Yu et al. (2021) found that upstream tide-induced residual currents induced the upstream transport of freshwater around the Yellow River mouth. All these studies demonstrate that tides can influence the current or mixing processes and thus, modulate river water behavior over the shelf. However, most of these studies have focused on the effect of tides on river water transport in estuaries and their adjacent seas, while few studies have evaluated the effect of tides on river water behavior at the spatial scale of the entire shelf sea.

The eastern shelf seas of China (ESSC) with energetic tides have wide continental shelves (Figure 1). The ESSC includes the Bohai, Yellow, and the East China Seas. It connects the rivers of China and the Korean Peninsula to the Western Pacific Ocean. Among these rivers, the Yellow, Yalujiang, and Changjiang Rivers are the three with the largest discharge. The

Changjiang River, located on the west coast of the East China Sea, has an annual mean runoff of ~30,000 m³/s, accounting for more than three-quarters of the total amount of river runoff discharged to the ESSC (Liu et al., 2010). The Yalujiang and Yellow Rivers are located on the northern coast of the Yellow Sea and the southern coast of the Bohai Sea, with annual runoffs of ~1,000 m³/s and ~650 m³/s, respectively. Terrigenous materials from the three rivers are discharged into the ESSC, playing an important role in its ecosystem and biogeochemical cycling (Liu et al., 2010; Yang et al., 2018; Ding et al., 2020).

As mentioned previously, several studies have investigated the characteristics of water transport in the three rivers and highlighted the role of tides in river water behavior (Wu et al., 2011; 2014; Liu et al., 2012; Yu et al., 2021). However, they commonly focused on only one aspect of the river water behavior (i.e., either the transport direction, the pattern of the river plume, or transport timescale) at the spatial scale of the estuary or its adjacent area. The present study aimed at a comprehensive understanding of the effects of tides on the river water behavior over the entire shelf and selected the Yellow,

Yalujiang, and Changjiang Rivers located in the Bohai, Yellow, and East China Seas, respectively, as the study objects. Using numerical modeling and sensitivity experiments, this study assessed the effect of tides on river water behavior in the ESSC from the perspectives of transport pathways, transport timescales, and water concentration distribution.

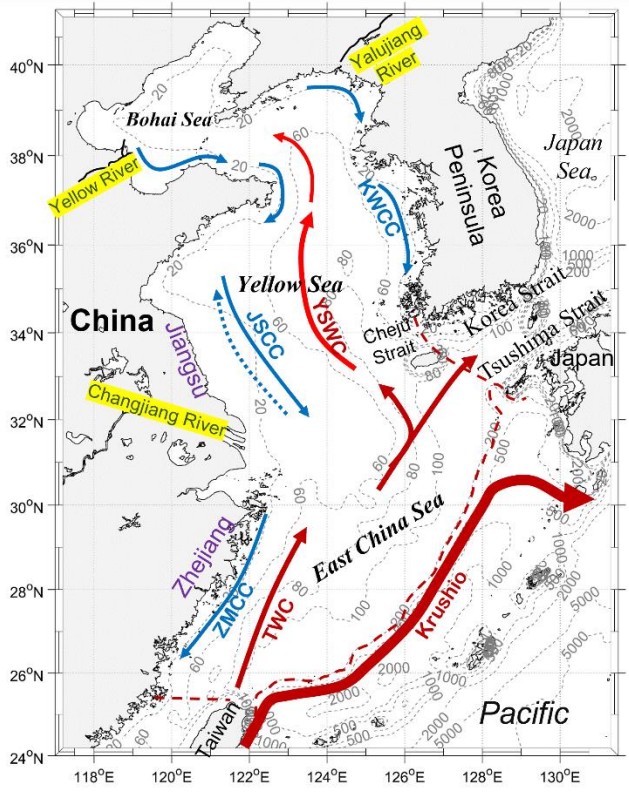

**Figure 1. Model domain, locations of the Yellow, Yalujiang, and Changjiang Rivers, and the primary shelf circulation in the**
**eastern shelf seas of China. The red and blue arrows denote the primary warm currents (TWC: Taiwan Warm Current; YSWC: Yellow Sea Warm Current) and coastal currents (KWCC: Korea West Coastal Current; JSCC: Jiangsu Coastal Current; ZMCC: Zhe-Min Coastal current), respectively. The blue dashed arrow denotes the JSCC in summer. The red dashed lines are the open boundary for the calculation of the transit time (see section 2.4).**

The remainder of this study is organized as follows: Section 2 introduces the models and diagnostic methods for river water transport pathways, timescales, and concentration. Section 3 presents the results of the impact of tides on the water behavior of the three rivers over the shelf. By analysing the change in the shelf circulation and the results of sensitivity experiments, Section 4 discusses the dynamic mechanism on the tidal effect on the river water behavior. The potential effect of the absence of tides on climate modelling are also discussed in Section 4. Finally, Section 5 provides a brief conclusion of the study.

## 2 Models and Methods

### 2.1 Hydrodynamic Model

The hydrodynamic model used in this study was the Princeton Ocean Model (Blumberg and Mellor, 1987). The model for ESSC was initially set up by Guo et al. (2003) and Wang et al. (2008). The model included four major tidal constituents ($M_2$, $S_2$, $K_1$, and $O_1$) and covered the entire ESSC (Figure 1). Its horizontal grid resolution was ~5-6 km. The model had 21 non-uniform sigma layers in the vertical direction with finer resolution in the upper layer. This study focused on the climatological state results, and thus the model was driven by climatological mean forcings, including winds, heat fluxes, precipitation, and evaporation (Wang et al., 2008). The monthly mean river runoff values of the Changjiang, Yellow, and Yalujiang Rivers (Table 1) were derived from measurements of river hydrological stations and the marine atlas, and were linearly interpolated to each time step during the model run. This model has been successfully applied to simulation studies of riverine nutrient transport and cycling in the ESSC (Zhao and Guo, 2011; Zhang et al., 2019; Wang et al., 2019; Zhang et al., 2021), the Yellow River age and plume in the Bohai Sea (Liu et al., 2012; Yu et al., 2020), the Yellow Sea cold water mass (Zhu et al., 2018), and the water residence time of the shelf sea (Lin et al., 2020). These studies have well validated the model and demonstrated the model reliability and applicability in the ESSC. The readers are referred to Wang et al. (2008) and Guo et al. (2003) for more detailed descriptions. The hydrodynamic fields from the hydrodynamic model, including the sea level, water depth, velocity, and diffusivity coefficients with a time interval of 0.5 hours were used to drive the particle-tracking and passive tracer models as both the particle-tracking and tracer models were decoupled from the hydrodynamic model. It should be noted that though surface evaporation was considered in the hydrodynamics model which could be the surface sink for the river water, that was not considered in the particle-tracking and tracer models.

Table 1. The monthly mean river runoff ($m^3$/s) of the Changjiang, Yellow, and Yalujiang Rivers in the hydrodynamic model.

| Rivers | Jan. | Feb. | Mar. | Apr. | May. | Jun. | Jul. | Aug. | Sep. | Oct. | Nov. | Dec. |
|---|---|---|---|---|---|---|---|---|---|---|---|---|
| Changjiang | 10834 | 12017 | 17171 | 25282 | 33188 | 39942 | 52840 | 44952 | 40323 | 32117 | 22136 | 13776 |

| Yalujiang | 656 | 657 | 709 | 693 | 695 | 828 | 1353 | 2117 | 1158 | 699 | 678 | 685 |
| Yellow | 378 | 296 | 299 | 209 | 216 | 271 | 907 | 1481 | 1345 | 1084 | 619 | 440 |

## 2.2 Particle-tracking and passive tracer models for diagnosing river water behavior

In this study, the particle-tracking method used to obtain the trajectory of river water over the shelf is from the estuarine and coastal ocean model coupled with a sediment transport module (ECOMSED) (Blumberg, 2002), which has been used in the study of the Yellow River water transport in the Bohai Sea (Liu et al., 2012; Wang et al., 2013). In the numerical experiments, we released 1000 particles at each of the three river boundaries at 0 o'clock every day and ran the particle-tracking model for 30 years. The location of each particle and its time after release were recorded for the analysis of the particle trajectory and calculation of the river water age.

To quantify the concentration of river water in the shelf seas, three passive tracers were released at the three river boundaries with a dimensionless concentration of 1. Thus, the passive tracers of each river were equivalent to the freshwater concentration, that is, the tracer concentration was 1 for freshwater and 0 for pure seawater. Tracer concentrations were calculated using an offline advection-diffusion module from the Marine Environment Research and Forecasting model (MERF) (Liu et al., 2016; Lin and Liu, 2019a; Tang et al., 2021), in which the TVDal and central-difference algorithms were used in the discretization of the advection and diffusion terms, respectively (Lin and Liu, 2019b). The tracer model ran for 30 years and the results of the last year were outputted and analyzed.

## 2.3 Characterizing the pathway and transit area of river water

Based on the results of the particle-tracking model, the emergence probability of the particles at each grid over the shelf was used to characterize the river water pathways and was calculated at $0.5° \times 0.5°$ grid cells by dividing the vertically integrated number of particles emerging in the grid cell by the total number of particles released per day (i.e., one thousand). A particle can re-enter a grid more than once, in which case the same particle is counted only once for a given grid. In this way, we obtained the results of the emergence probability for particles released on each day of one year. The annual mean emergence probability was used in the analysis of the river water transport pathway, which was calculated by averaging the emergence probability on all days of the year. The main water transport pathway is indicated by locations with high particle emergence probabilities.

Then, the transit area of the river water was calculated as the particles' emergence probability in each grid multiplied by the area of the corresponding grid and summed over the shelf to quantify the magnitude of the influence range of the river on shelf seas. A larger (smaller) transit area indicates more dispersed (concentrated) transport of the river water.

## 2.4 Transport timescales of river water

Water age and transit time were used to quantify the transport timescales of river water over the shelf.

### 2.4.1 Water age

Water age is defined as the time elapsed since a river water particle enters the domain of interest (Deleersnijder, 2001), and can quantify the transport rate of the river water over the shelf. For a particle released at time $t_0$, it moves to some position at time $t$, and the water age at this position is $t$-$t_0$. Then, the mean water age for each grid cell was calculated using a weighted

average of all the particle ages at a grid cell at time $t$. Once one particle had left the ESSC, it would no longer participate in the calculation of the mean water age. Because the same number of particles was released every day, particles released on different days denoted different water masses due to variation of the river discharge. Thus, the weighted average of the mean water age was calculated using the river discharge on the particle release day as the weight coefficient. The mean water age used in the analysis was calculated by averaging the particle age at $0.5° \times 0.5°$ grid cells. Due to space limitations, this study

only shows the annual mean of the vertical average results for the water age of the three rivers.

### 2.4.2 Transit time

The transit time of the river water over the shelf is defined as the time the river water particle spends from the river boundary to the shelf boundary (red dashed lines in Figure 1), which can quantify the total retention time of the river water in the shelf seas. A river particle released at time $t_0$ leaves for the first time the shelf at time $t_1$, and its transit time, according to the

definition is $t_1$-$t_0$. The re-entry process was not considered in the calculation of the transit time and water age.

### 2.5 Physical meanings of the indicators

Overall, five indicators (i.e., the emerging probability, transit area, water age, transit time, and tracer concentration) were used to characterize the behavior of the river waters. The emerging probability of the particles reflects the proportion of river water that passes through the region, which is mainly related to the shelf currents. The regions with higher emerging

probability indicate a greater amount of river water passing through them and thus are the main pathway of river water transport. The transit area can be an approximate representation of the area through which the river flows. Water age reflects how long the river water has been on the shelf after leaving the estuary, while the transit time reflects the total time the river water takes from entering to leaving the shelf. The tracer concentration reflects the stock of the river water over the shelf, which can influence the salinity field. It should be noted that the magnitudes of the emerging probability, water age, and

tracer concentration may not be directly related. For instance, a small proportion of one river water may stay in region A for a long time while the most proportion of the river water may quickly leave the shelf through region B. Then, region A would obtain a relatively large water age but a relatively small emerging probability compared with region B. However, the tracer

concentration in region A could be either high or low which is determined by the accumulated amount of river water in this region.

## 2.6 Sensitivity experiments

Numerical experiments were designed to explore the effect of tides on the river water fate in the ESSC, which were conducted by using the model run with tides (hereafter termed 'Control run') and the model run without tides (hereafter termed 'No-Tide case'), respectively. In both experiments, the hydrodynamic model was run for three years which was long enough for the model to get steady seasonal cycles (see Figure S1 in the supplementary material), and the model data output of the last year was used for analyzing and driving the particle-tracking and passive tracer models. Then, the water transport pathways and timescales and the water concentrations of the rivers in the ESSC were determined using the same procedures described in Sections 2.3 and 2.4.

## 3 Results

By comparing the model's Control run and No-Tide case results, the effect of tides on the behavior of the three rivers' waters in the ESSC was analyzed for three aspects: the effect on river water transport pathways, transport timescales, and concentrations.

### 3.1 Effect on transport pathways

Significant differences in the water transport pathways of the three rivers between the Control run and No-Tide case were observed (Figure 2). As shown in the results of the Control run, most of the Yellow River water passes through the Bohai Strait, enters the Yellow Sea, and finally leaves through the Cheju Strait (Figure 2a). This emergence probability suggests that the Yellow River water passes through most of the Yellow Sea. The Yalujiang River water passes mainly through the eastern Yellow Sea and leaves it through the Cheju Strait (Figure 2b). The Changjiang River water flows northeastward after leaving the Changjiang estuary and leaves China's eastern shelf through the Korea-Tsushima strait, with a probability of ~10–20 % of it passing through the Yellow Sea (Figure 2c). The Changjiang River water is concentrated in the South Yellow Sea and north of the East China Sea. In addition, the emerging probability for the Changjiang River showed spatial discontinuity (Figure 2c), which could be due to the northward transport of the river water by the Yellow Sea Warm Current. The norward transport brought part of the Changjiang River into the Yellow Sea and thus rapidly reduced the emerging probability of the river water in the middle of the Yellow Sea.

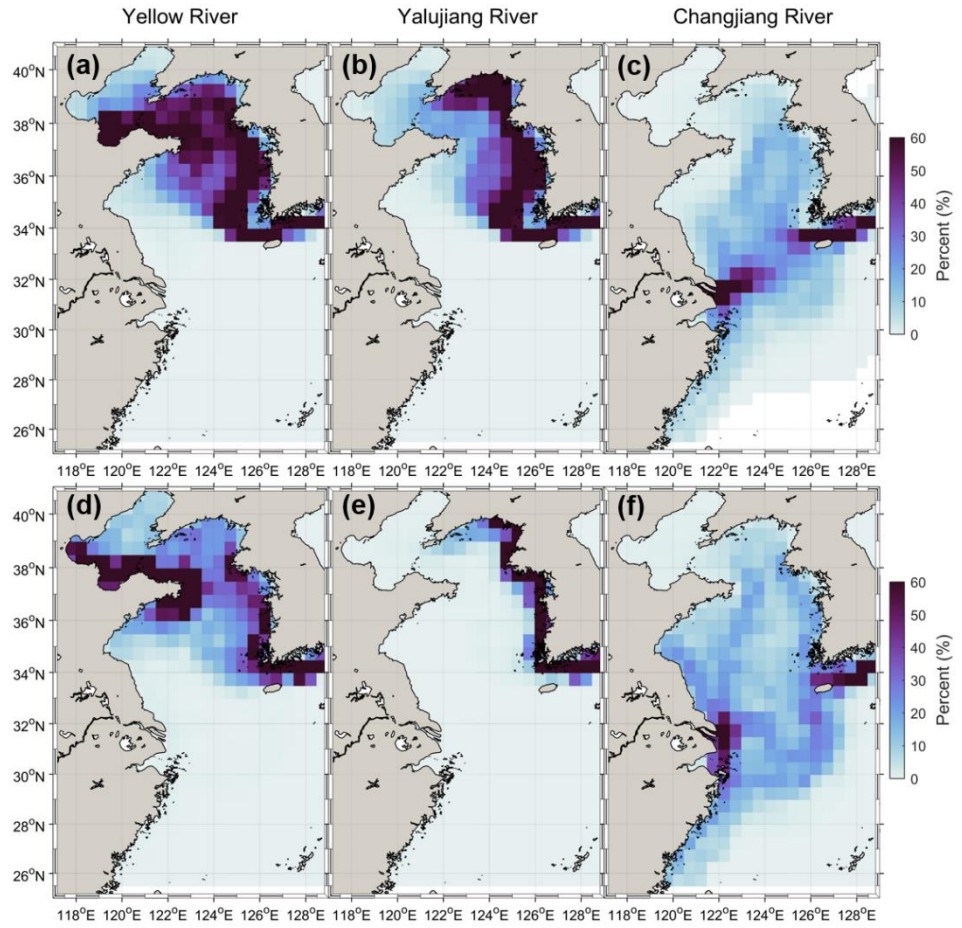

Yellow River        Yalujiang River        Changjiang River

**Figure 2. The annually averaged emergence probability of the particles released at the three rivers' boundaries for the Control run (a–c) and No-Tide case (d–f), which are indicated by the emergence probability of the particles in the grids (in percent).**

In comparison to the Control run, the water transport pathways of the Yellow River and Yalujiang River for the No-Tide case were more concentrated in the western coastal region of the Yellow Sea, as shown in the map of their emergence probability (Figures 2d and 2e). However, the emergence probability of the Changjiang River water for the No-Tide case increased in the Yellow Sea and East China Sea, particularly at the Jiangsu and Zhemin Coasts (Figure 2f). In comparison, the transit areas of the Yellow, Yalujiang, and Changjiang Rivers for the Control run are 2556, 2027, and 1537 km², respectively, while they are 2224, 1214, and 1851 km², respectively, for the No-Tide case. As compared with the control run, the transit areas of the Yellow and Yalujiang Rivers for the No-Tide case decreased by 13 % and 40 %, respectively, while that of the Changjiang River increased by 21 %. The above results suggest that the tidal effect induced more dispersed transport for the Yellow and Yalujiang Rivers' waters but more concentrated transport for the Changjiang River water.

### 3.2 Effect on transport timescales

The transport timescales of the river water for the No-Tide case were significantly decreased as against the Control run. The water age for the three rivers showed high values (4–10 years) in the Bohai Sea and the Yellow Sea and much lower values (less than 1 year) in the East China Sea (Figures 3a–3c). The mean age of the Yellow and Yalujiang Rivers water was 4.7 and 4.5 years in the Bohai Sea and 5.5 and 3.1 years in the Yellow Sea, respectively. The water age of the Changjiang River in the Yellow and East China Seas was 1.3 and 0.3 years, respectively. Moreover, the relatively high water age of the three rivers is concentrated in the central Yellow Sea (123–126 E°, 34–39 N°), where the Yellow Sea Cold Water Mass (YSCWM) occurs every summer, suggesting that the YSCWM region could trap river water and terrigenous materials for several years. However, the emergence possibility in the YSCWM region was relatively low (Figure 2), as only a small proportion of the river water was trapped in the region of the YSCWM and most of the river waters were transported along the west coast of the Korean peninsula. The YSCWM occurred at the bottom of the central Yellow Sea during the stratified season. It formed mainly due to the retention of the winter cold water (Zhang et al., 2008), while the water in winter is well mixed in the Yellow Sea (e.g., Zhu et al., 2018; Lin et al., 2019). Thus, the river water trapped by the YSCWM during the stratified season should be mainly from the river water in the Yellow Sea in winter.

In comparison, when the tides were removed, the rivers' water age decreased significantly (Figures 3e–3f). For the No-Tide Case, the average age of the Yellow and Yalujiang Rivers water was 1.6 and 0.5 years in the Bohai Sea and 1.0 and 0.3 years in the Yellow Sea, respectively, and decreased by averagely more than 80 % as compared with the Control run. The mean water age of the Changjiang River in the Yellow Sea decreased to 0.5 years (by ~60 %) in the No-Tide case, while in the East China Sea it remained basically unchanged. The dramatic decrease in the river water age for the No-Tide case suggests that the tides could significantly slow down the transport of river water over the shelf, especially in the relatively enclosed Yellow and Bohai Seas.

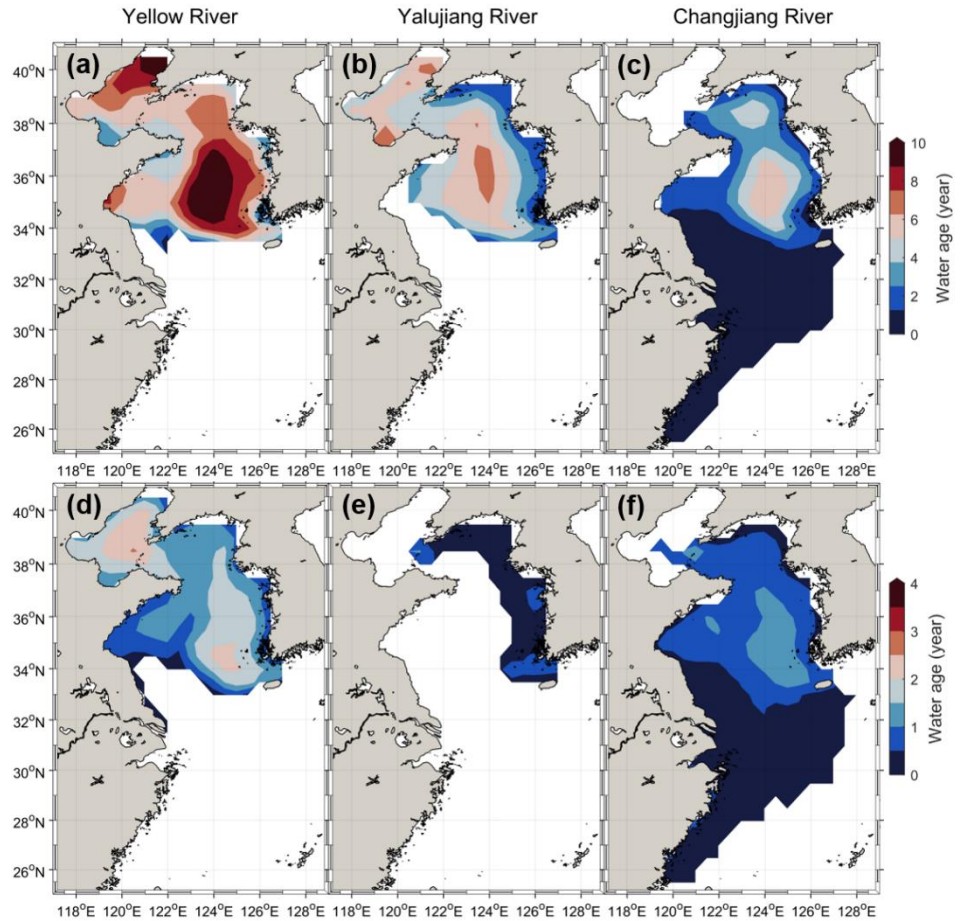

**Figure 3. The annually and vertically averaged water age of the three rivers for the Control run (a–c) and No-Tide case (d–f). Only results at locations with particle emergence probability of > 1 % have been presented.**

The shorter transport timescales of the rivers' waters for the No-Tide case are also shown in the transit time results (Figure 4). The annual mean transit time of the Yellow, Yalujiang, and Changjiang Rivers are 9.4, 5.6, and 2.2 years, respectively, while they decreased to 2.4, 0.4, and 1.2 years, respectively, in the No-Tide case. The apparent decrease in transit time indicated a much shorter retention in the shelf seas and a faster export rate of river water from the shelf seas for the No-Tide model than for the Control model considering tides. The mean transit time of the waters of the Yellow and Yalujiang Rivers for the No-Tide case decreased by 75 and 93 %, respectively, which are larger than that of the Changjiang River (45 %), suggesting that the effects of tides on the river water export rate could be more strong for rivers in relatively enclosed seas than those in relatively open seas. Moreover, as against the Control run, the river water transit time for the No-Tide case showed a different seasonal variation pattern and a decreased variance.

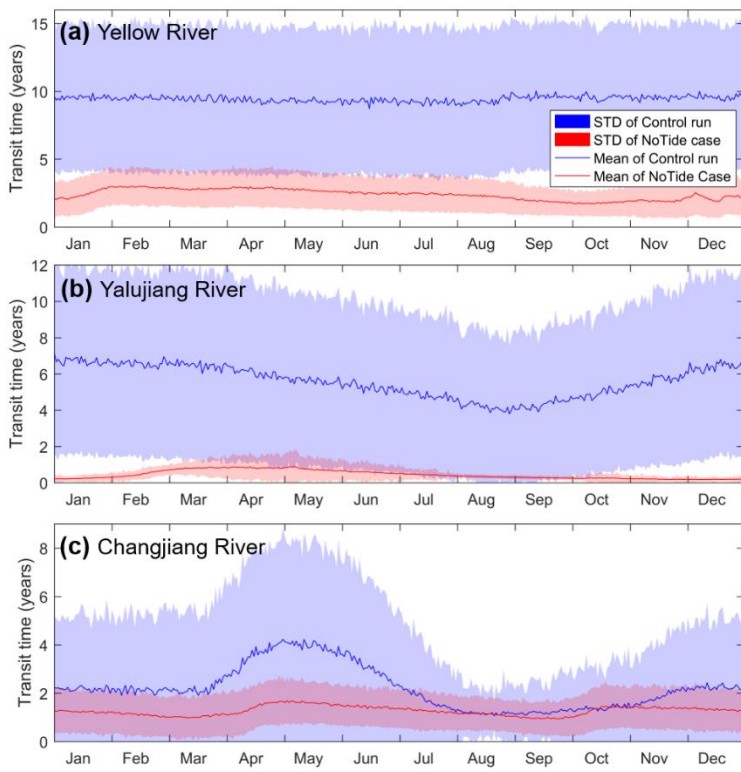

**Figure 4. The daily mean water transit time over the shelf of the three rivers for the Control run (blue) and No-Tide case (red). STD in the figure label denotes the standard deviation of the transit time for particles released on the same day.**

### 3.3 Effect on river water concentrations

Given their effect on river water pathways and timescales, it is expected that tides would influence the river water concentrations over the shelf. As indicated in Figure 5, the water concentrations of the three rivers in the No-Tide case were approximately one order lower in magnitude than those in the Control run, especially in the Yellow and Bohai Seas. As

against the Control run, when the tides were excluded, the annual mean river water concentrations of the Yellow, Yalujiang, and Changjiang Rivers over the entire shelf were reduced by 73, 84, and 76 %, respectively. The mean reduction in the water concentration of the three rivers in the Bohai and Yellow Seas is over 75 %, which is several times more than that in the East China Sea, suggesting that the tidal effect on river water concentrations is more significant in relatively enclosed seas than in relatively open seas. In addition, relatively low tracer concentrations but relatively high emerging probabilities occurred in

Cheju Strait (Figures 5 and 2). This is because when the river water was transported to the Cheju Strait, it had been mixed with shelf seawater which decreased the river water concentration. However, since the Cheju Strait is an important outlet for river water, most of the river particles would pass through the strait which induced the high emerging probabilities of the river water at the Cheju Strait.

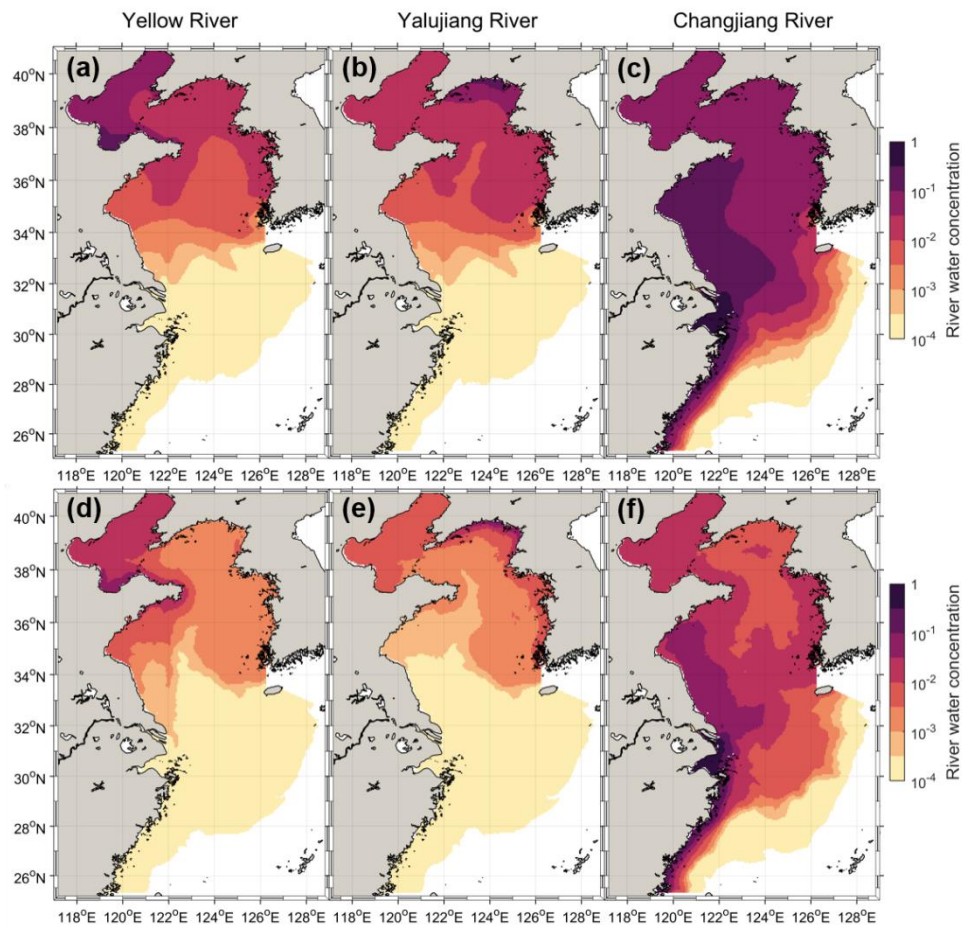

**Figure 5. The annually and vertically averaged river water concentrations of the three rivers for the Control run (a–c) and No-Tide case (d–f) (The maps show the logarithmic distribution of the obtained river water concentrations).**

A change in river water concentration can directly induce a change in seawater salinity over the shelf. As shown in Figure 6, the salinity in the estuaries and coastal waters is lower than 30 because of the input of freshwater from the rivers, while the

265 salinity in the East China Sea is relatively high (>33) because of the salty water input from the Kuroshio Current (Figure 6a). The seawater salinity in the No-Tide case increased significantly as against the Control run, particularly in the areas adjacent to the estuaries and in the Bohai and Yellow Seas (Figures 6b and 6c). This can be understood as the faster export of the river water in the No-Tide case (Figures 3 and 4) leaving a smaller proportion of river freshwater in the shelf sea (Figure 5) and resulting in higher salinity over the shelf. The increase in salinity was more than 3 in the estuaries of the Yellow,

Yalujiang, and Changjiang Rivers and more than 1 (on average) in the Yellow and Bohai Seas (Figure 6c), while in the East China Sea, the magnitude of salinity change was smaller than 0.5, suggesting that the tidal effect on seawater salinity is more significant in relatively enclosed seas than in relatively open seas.

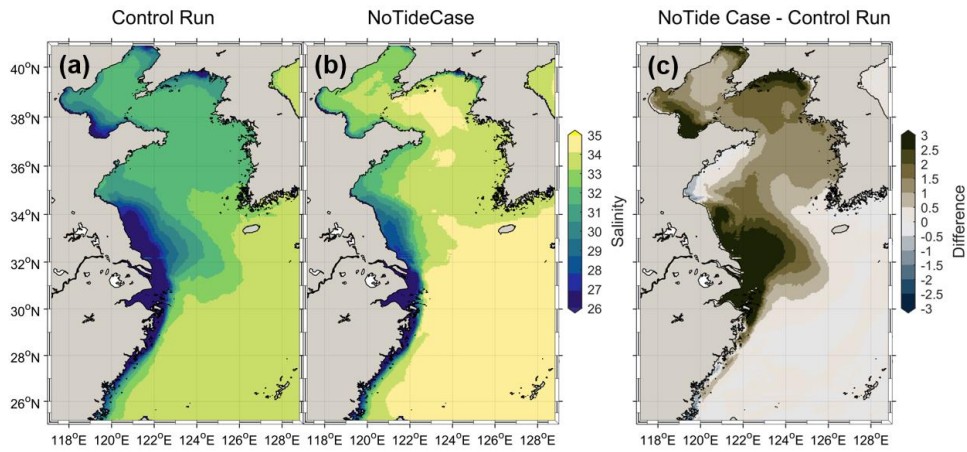

**Figure 6.** The annually and vertically averaged salinity of the shelf water for the Control run (a), No-Tide case (b), and their difference (c). Positive (negative) values denote the salinity of the No-Tide case as being higher (lower) than that of the Control run.

## 4 Discussion

### 4.1 Role of tidally induced change in the mixing and advection on the river water behavior

Changes in the river water transport pathways and timescales, and thus the river water concentrations and seawater salinity in the shelf seas, indicate a significant effect of tides on the behavior of the river water over the shelf. The river water behavior over the shelf is determined by two processes, namely, advection and mixing. Tides can affect shelf current and water mixing and thus, influence river water transport over the shelf (Moon et al., 2009; Palma et al., 2004; Lin et al., 2020; Wang et al., 2013; Wu et al., 2018). To assess the respective effects of the changes in mixing and currents (changes in shelf currents and mixing are shown in Section 4.2 and in Figure S2 in the supplementary material, respectively), induced by tides on river water transport, we designed two additional numerical experiments similar to the No-Tide case, except that in one, the diffusivity coefficients driving the particle-tracking and passive tracer models were replaced by those in the Control run (hereafter termed "Tidal-Mixing case"), and in the other, the velocity and sea level driving the particle-tracking and tracer transport models were replaced by those in the Control run (hereafter termed "Tidal-Advection case"). The effect of the tidally induced change in the mixing and current on river water transport could be identified by comparing the results of the Tidal-Advection and Tidal-Mixing cases with those of the No-Tide case and Control run.

The river water transport pathways in the Tidal-Advection and Tidal-Mixing cases show a similar result to that in the Control run and No-Tide case, respectively (Figure 7), suggesting that tidally induced changes in the shelf currents are more important than those induced by water mixing for the river water pathways in the shelf sea. This is reasonable because the currents usually dominate horizontal transport, whereas mixing influences vertical transport.

For the transport timescales, the water age of the Tidal-Advection case was much higher than that of the No-Tide case and even higher than that of the Control run, whereas the water age of the Tidal-Mixing case was slightly lower than that of the No-Tide case (Figure 8). Similar to the results of the water age, the annual mean transit times of the Tidal-Mixing case were 2.4, 0.65, and 1.0 years for the Yellow, Yalujiang, and Changjiang Rivers, respectively, which are very close to the results of the No-Tide case (Figure 9); however, the transit time of the Tidal-Advection case was much higher than that of the No-Tide

case. The water age and transit time results suggest that the tidally induced changes in currents and mixing had opposite effects on the river water transport timescale, which decreased and increased the river water transport rate, respectively. As the transport timescales of the Control run (with tides) were much higher than those of the No-Tide case, we can infer that the change in shelf currents induced by tides plays a dominant role in the transport rate of river water over the shelf.

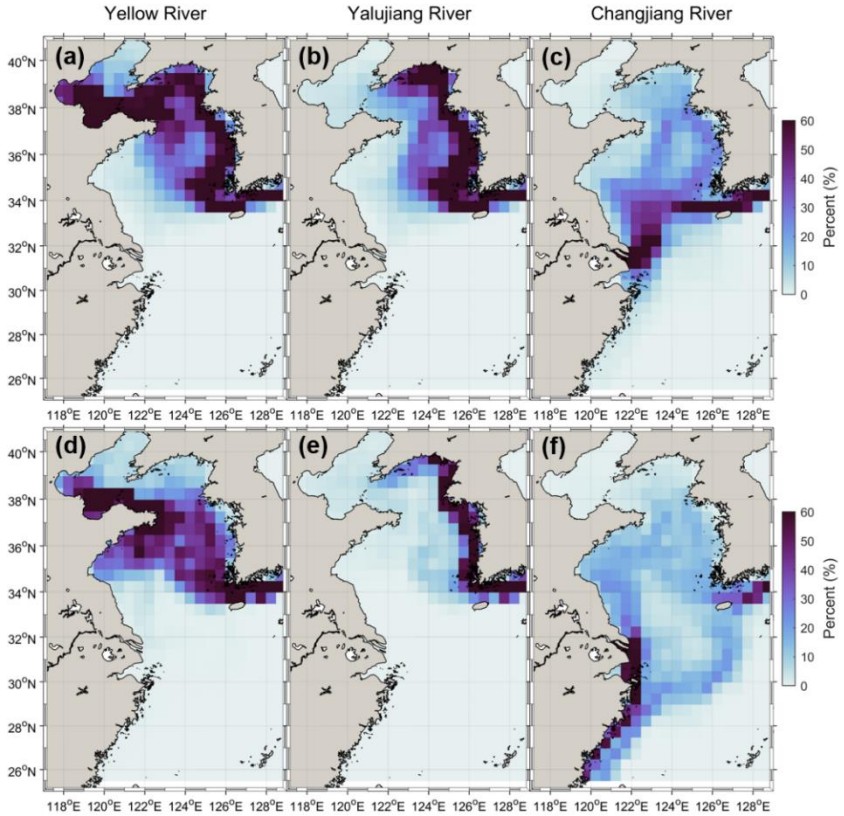

**Figure 7. The annually averaged water transport pathways of the three rivers over the shelf for the Tidal-Advection case (a–c) and Tidal-Mixing case (d–f), indicated by the emergence probability of the particles in the grids (in percent).**

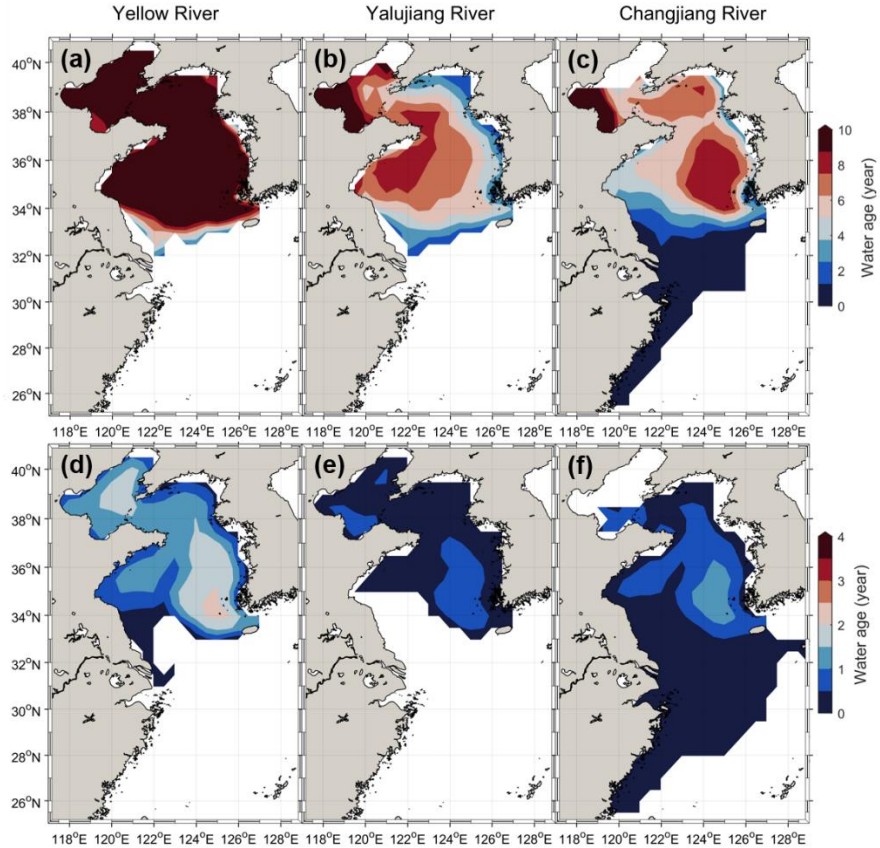

**Figure 8. The annually and vertically averaged water age of the three rivers for the Tidal-Advection case (a–c) and Tidal-Mixing case (d–f).**

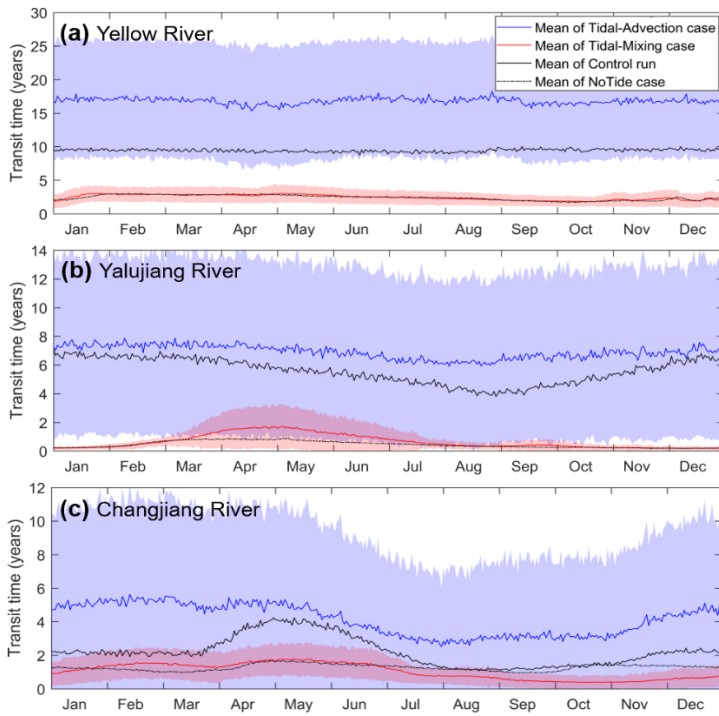

**Figure 9. The daily mean water transit time over the shelf of the three rivers for the Tidal-Advection case (blue) and Tidal-Mixing case (red), along with the Control run (black) and No-Tide case (dotted black).**

### 4.2 Tidally-induced changes in the shelf current and its effect on river water transport

To further understand the effect of tidally induced changes in currents on river water transport, we compared and analyzed the shelf currents of the Control run and the No-Tide case (Figure 10). When the tides were removed, the change in the shelf currents mainly occurred in the coastal region and the central Yellow Sea (Figures 10a-10d), and the change in winter was more significant than that in summer (Figures 10e-10f and Figure S3 in the supplementary material). During winter, the coastal currents along the Jiangsu and west Korea in the No-Tide case were several times stronger as against those in the

Control run (see Figures 10d). The Yellow Sea Warm Current (YSWC) was also significantly strengthened in the model without tides (Figure 10b). The analysis of water transport fluxes by Lin et al. (2020) showed the winter season accounted for ~73% of the volume transport for the entire year and suggested that the winter processes dominate the water exchange of the Yellow Sea. Due to the strong surface cooling, the coastal water (even in the central Yellow Sea) in winter was well-mixed. Thus, the change in the barotropic process could dominate the change of the shelf currents. In addition, the change in

the coastal current along the Jiangsu and Zhejiang coasts showed southward in winter and northward in summer (Figures 10e and 10f), which are consistent with the directions of the seasonal wind-driven currents (northerly wind in winter and southerly wind in summer). This consistency indicates that the wind-driven coastal current was significantly weakened by

tides, which can be explained by the nonlinear interaction between tides and wind-driven current in a quadratic bottom friction term (e.g., Moon, 2005; Wu et al., 2018). When considering tides, the period-mean bottom resistance becomes about one order larger due to the quadratic nature of the bottom friction and thus greatly reduced the wind-driven coastal currents (Wu et al., 2018).

The relatively weak YSWC in the Control run is mainly related to the weakened coastal current and enhanced bottom friction. Several studies have interpreted the formation of the YSWC using the upwind theory (e.g., Hsueh & Yuan, 1997; Isobe, 2008; Lin et al., 2011). In the semi-enclosed Yellow Sea, the wind stress is the dominant forcing in winter and thus the currents along the eastern and western coasts are in the same direction as the northerly wind. The southward coastal currents on both sides can reduce the water elevation inside, pile up water mass at the downwind end, and form the negative pressure gradient (decrease to the north). Along the deep trough of the central Yellow Sea, the depth-averaged wind stress is small and too weak to counter the northward pressure gradient force. This imbalance results in a pressure-driven flow that is opposite to the direction of the surface wind along the trough of the Yellow Sea. Under the effect of the Coriolis force, the upwind current is further geostrophically adjusted and slightly shifted to the western side of the trough to satisfy the balance of the potential vorticity (Lin et al., 2011). When considering tides, enhanced bottom friction weakened the coastal currents and thus decreased the water elevation difference and thus the pressure gradient (Moon 2005; Lin et al., 2020). Meanwhile, the enhanced bottom friction below the YSWC further countered a part of the northward momentum (Moon 2005). Thus, the weaker YSWC was presented in the model with tides.

Besides the tidally enhanced bottom resistance, the tidally enhanced mixing and induced residual current and residual transport could also modulate the shelf currents, especially during the stratified season (e.g., Moon, 2005; Lie and Cho, 2016; Moon et al., 2009; Wu et al., 2011; Wu et al., 2014; Wu & Wu, 2018). Moon et al (2009) suggested that the tidal forcing induces a strong southward residual flow along the western slope of the Yellow Sea in summer, which can explain the change of northward current in the NoTide case in the middle of the Yellow Sea (Figure 10f). Wu et al. (2011) suggested that tidally enhanced mixing resulted in a strong horizontal salinity gradient at the north of the Changjiang River mouth, which acted as a dynamic barrier and restricts the northward transport of the Changjiang River water to the Jiangsu coast in summer. In addition, strong tidal mixing induced a mixing front along the Zhe-Min coast and maintained a down-shelf frontal current (Wu & Wu, 2018), which was important for the southward transport of the Changjiang River water. Overall, the mechanism of the tidal effect on the shelf current is complicated and diverse. However, the tidally enhanced bottom friction should be the dominant mechanism due to the important role of winter currents in the water exchange and the significant change in the winter coastal current in the Control run and NoTide case.

In contrast to the Bohai and Yellow Seas, there was a relatively small difference in the shelf currents in the central East China Sea between the Control run and No-Tide Case (Figure 10d). The different effects of tides on the Bohai and Yellow Seas and the East China Sea could be related to both the geometry and the water depth, as both of them can influence the water exchange and the strength of the tidal current. The semi-enclosed geometry in the Bohai and Yellow Seas induced relatively weak water exchange compared to the relatively open East China Sea. Thus, the coastal currents are important

dynamics for the water exchange in the Bohai and Yellow Seas and the tidal effect on the coastal currents significantly influenced the river water behavior in the Bohai and Yellow Seas. On the other hand, the change in the bottom friction induced by the tides is an important mechanism for the tidal effect on the shelf currents. Due to the larger inertia and stronger stratification, the shelf currents in relatively deep water could be less influenced by changes in bottom friction than that in relatively shallow water. Thus, the effect of tides on the shelf current and river water concentration in the East China Sea was relatively smaller than those in the Yellow and Bohai seas.


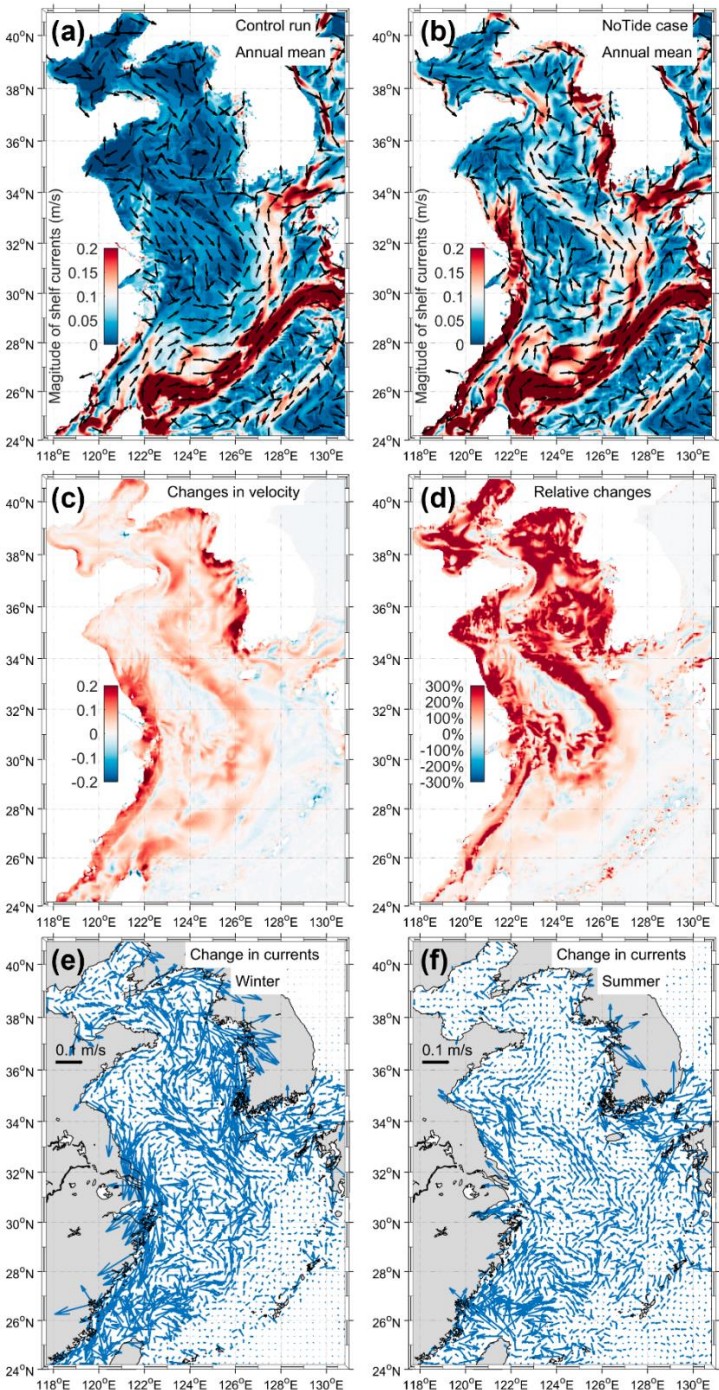

**Figure 10. Comparison of the shelf currents between the Control run and NoTide case. (a) and (b) are the annual mean and vertically averaged velocities for the Control run and No-Tide case, respectively. The arrow and color in (a) and (b) denote the direction and magnitude of the velocity, respectively. (c) and (d) show the change and relative change in the magnitude of the shelf currents after removing tides in the model. (e) and (f) show the changes in shelf currents in February and August, respectively.**

The changes in the shelf currents further changed the behaivor of the river water over the shelf. For the Yellow and Yalujiang Rivers, the significantly intensified coastal currents in the Yellow and Bohai Seas in the No-Tide case accelerated water transport along the coasts to the Cheju Strait and induced much smaller transport timescales than those of the Control run. Likewise, the enhanced coastal currents and shelf circulations of the No-Tide case can accelerate river water transport over the shelf and thus induce much shorter water transport timescales in the Changjiang River. However, the tidal effects on

the pathways of the Yellow and Yalujiang Rivers and the Changjiang River were different. Tides induced more dispersed transport for the water of the Yellow and Yalujiang rivers, but more concentrated transport for the Changjiang River water (Figure 2). The different tidal effects should be related to the different water transport patterns of the rivers (Figure 11). For the Yellow and Yalujiang Rivers water, the west coast of the Korea Peninsula with strong tides and cross-shelf tidal currents (see the tidal current ellipses and cotidal map in Figure S4 in the supplementary material) was their main transport pathway.

Without considering tides, the southward coastal current along the west coast of the Korean Peninsula was intensified, especially in winter, which accelerated the river water export and reduced the transport timescales of the Yellow and Yalujiang Rivers' waters. When considering the tides, on one hand, the cross-shelf tidal currents along the coast could increase the cross-shelf water dispersion as the magnitude of tidal dispersion is proportional to the tidal velocity (Geyer & Signell, 1992). On the other hand, the weakened coast current slowed down the along-shore transport of the river waters and

further promoted cross-shelf water dispersion. Thus, compared to the NoTide case, the Control run obtained a more dispersed transport for the Yellow and Yalujiang Rivers' waters. For the Changjiang River, there are three major branches for the river water transport, i.e., the northeastward branch to Cheju Island, the northward branches to the Jiangsu coast, and the southward branch to the Zhejiang coast (Wu et al., 2014). The northeastward branch to Cheju Island is the dominant one for the Changjiang River water transport. In the NoTide case, the intensified coastal currents could increase the river water

transport to the Yellow Sea along the Jiangsu coast in summer and to the East China Sea along the Zhejiang coast in winter, which intensified the northward and southward branches of the Changjiang River water transport. In addition, the flow field at the offshore water of the Changjiang estuary in the Control run showed a pattern of a mesoscale eddy which might trap some river water and reduce the dispersion of the water. Thus, the water particles of the Changjiang River in the NoTide case were more dispersed than those in the Control run.

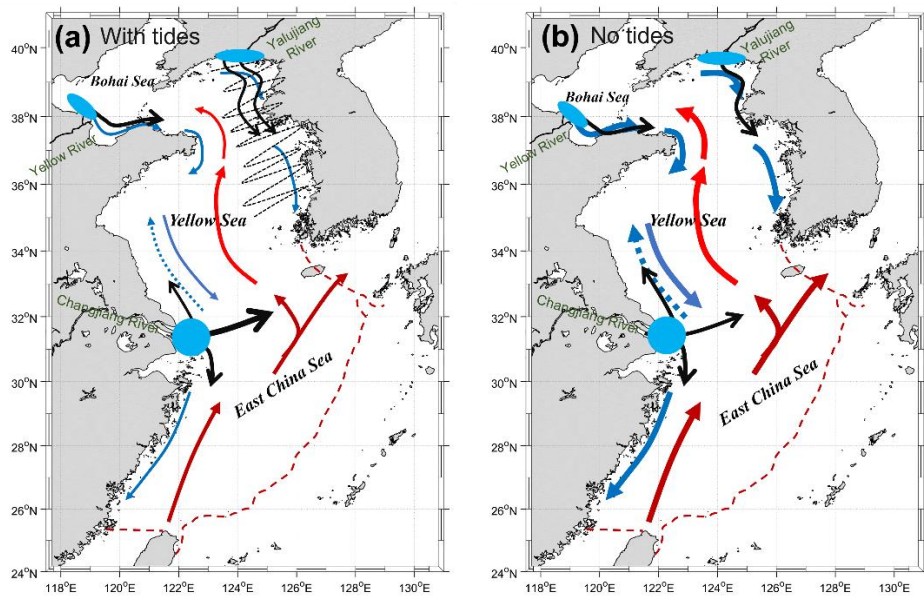


**Figure 11. Schematic map of the shelf currents and the major pattern of the river water transport for the Control run (a) and NoTide case (b). The blue and red arrows denote the coastal currents and warm currents, respectively. The blue dashed arrow at the Jiangsu Coast denotes the coastal current in summer. The thickness of the arrows denotes the intensity of the shelf currents. The black dashed wavy line in (a) denotes the cross-shelf dispersion induced by tides. The black arrows denote the direction and**
**the branches of the river water transport.**

The experiential results also suggested that, under the effect of tides, the transit time of the three rivers water in the control run showed a high standard deviation (STD) (Figure 4), which could be related to the strong water dispersion and the high spatial variation in the water residence time. As shown in Lin et al. (2022), there is significant spatial variation in the water
residence time (WRT) on the eastern shelf of China. The WRT can reach 15 years in the Bohai Sea, 4-10 years in the Yellow Sea, and about 1 year in the East China Sea. The particles released at the estuaries were continuously dispersed as they move under the effect of tidal dispersion and mixing. Dispersed particles can go into different seas and get significantly different residence times over the shelf, which can explain the high STD of the transit time. In addition, the seasonal variability of the transit time of the Changjiang River water could be related to the seasonal variation in the shelf currents. In spring, the part
of the Changjiang River water is prone to be transported into the Yellow Sea under the effect of the southerly wind and thus obtained a longer retention time over the shelf than in other seasons. Meanwhile, the large range of the residence time of the Yellow Sea induced the large STD in spring.

**4.3 Whether a tidal parameterization can improve the simulation in a no-tide model?**

The present study shows that tides could have a significant effect on river water transport, concentration, and seawater salinity in shelf seas by changing the bottom resistance and reducing the shelf currents. Tidal parameterization can include the tidal effect on hydrodynamics in a no-tide model to some extent. To examine the availability of tidal parameterization in no-tide models for the simulation of river water transport on the shelf, we added a parameterized-tide experiment (PMT-Tide case), wherein, linear-type bottom friction considering the tidal effect proposed by Lee et al. (2000) was applied to the

hydrodynamic model. The method has been used in studies on the shelf currents in the Yellow Sea and ESSC (Moon et al., 2009; Lin et al., 2020). The details of the parameterization were presented by Moon et al. (2009) and Lin et al. (2020). Then, using the hydrodynamic forcing of the parameterized-tide model, the water transport pathways and timescales, and the water concentrations of the rivers in the ESSC were diagnosed using the same procedures described in Sections 2.3 and 2.4.

The experimental results showed that the tidal parameterization could not significantly improve the simulation of the river

water transport pathways and the pattern of the river water age over the shelf. As compared with the Control run, the river water pathways of the Yellow and Yalujiang Rivers in the PMT-Tide case were still very concentrated in the coastal region, and that of the Changjiang River was more dispersed in the Yellow Sea (Figure 12). The transit time of the Yellow River for the PMT-Tide case was close to the result of the Control run (~0.2 years bias), while those of the Yalujiang and Changjiang Rivers still showed a large bias (more than 3.4 years) as compared to the Control run (Figure 13). When compared with the

No-Tide case, the PMT-Tide case could increase the water concentration of the Yellow and Changjiang Rivers (Figure 12) and thus improve the simulation of salinity over the shelf to some extent; however, there was still a large bias in the simulated salinity (> 3) in the estuarine zones (Figure 14). Therefore, the tidal parameterization in a no-tide model could not adequately consider the effect of tides on river water transport, implying that a better tidal parameterization for no-tide ocean models needs further development and examination in future work, or explicitly considering the tides in ocean models is

necessary.

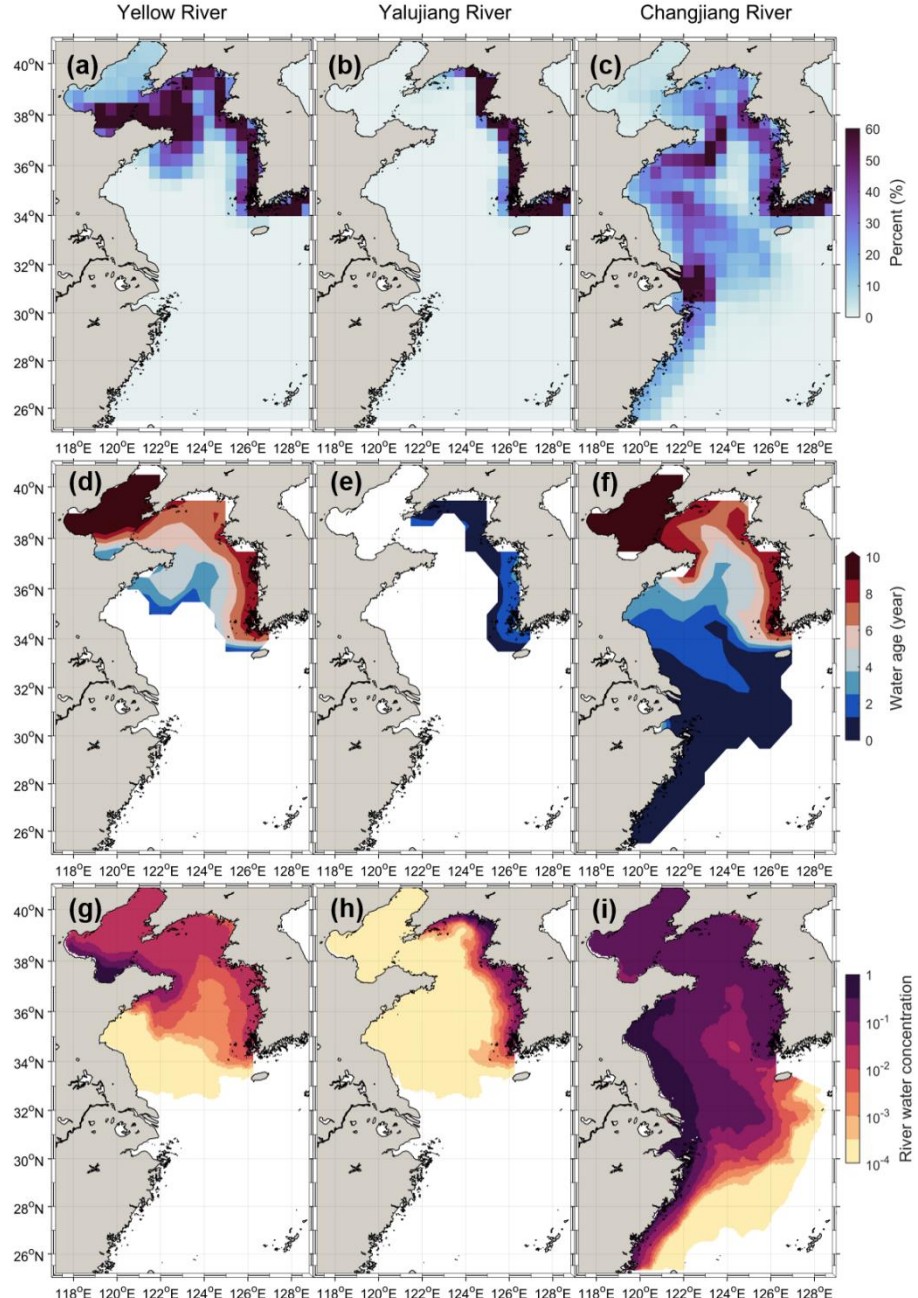

**Figure 12. The three rivers' waters transport pathways, age, and concentration for the PMT-Tide case. The annually averaged emerging probability of particles (a–c). The annually and vertically averaged water age (d–f). The annually and vertically averaged river water concentration (g–i).**


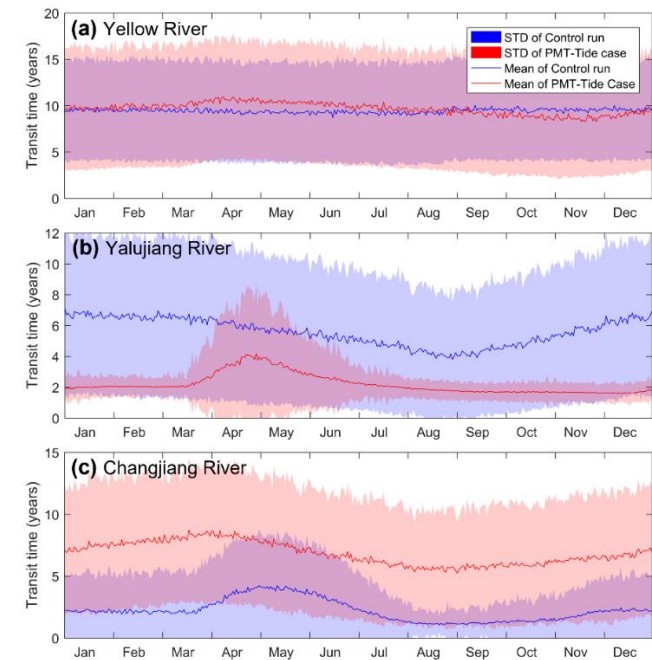

**Figure 13. The daily mean water transit time over the shelf of the three rivers for the Control run (blue) and PMT-Tide case (red). STD in the figure label denotes the standard deviation of the transit time for particles released on the same day.**

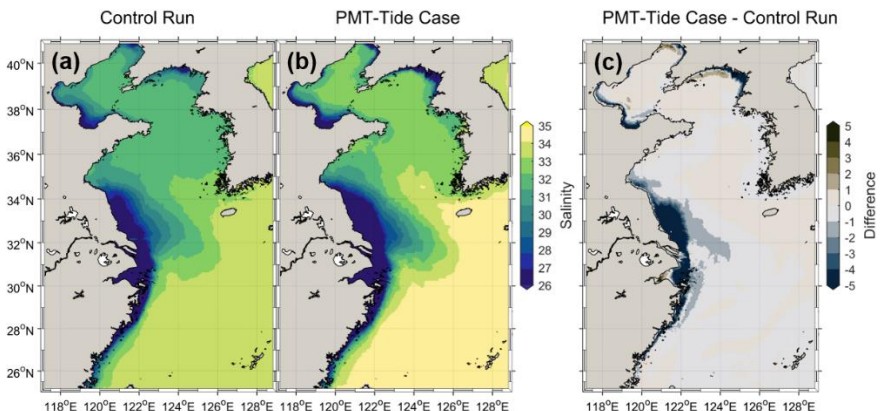

**Figure 14. The annually and vertically averaged salinity of the shelf water for the Control run (a), the PMT-Tide case (b), and**
**their difference (c). Positive (negative) values denote the salinity of the No-Tide case higher (lower) than that of the Control run.**

### 4.4 Implications on climate modeling

The present study demonstrated that tides have a significant effect on river water transport pathways, timescales, river water concentration, and shelf seawater salinity. Moreover, the improvement of tidal parameterization in the simulation of a no-tide model could be very limited. Therefore, if earth system models or climate models do not explicitly considered tides, the
model results will have a large bias in the river water transport process over shelf seas. Consequently, such biases may

seriously affect the accuracy of global model predictions of global water and carbon cycles and even climate change for the following reasons. First, river runoff which accounts for about 10% of the global meridional water fluxes play a significant role in the ocean circulation through affecting the water balance and salinity of the oceans (Oki et al., 1995; 1999). The absence of tides in global models could accelerate the freshwater export from shelf seas to deep oceans and influence the salinity balance and circulations in oceans. Second, river water is rich in terrigenous nutrients, which are an important source of nutrients for phytoplankton in shelf seas. Shelf seas contribute to approximately 1/3 of the marine primary productivity and 40 % of the ocean carbon storage, and the carbon pumping of shelf seas is an important way of global carbon sequestration, thus shelf seas playing a critical role in the global carbon cycle (Thomas, 2004; Laruelle et al., 2018; Dunne et al., 2007). The absence of tides will change the nutrient transport pathways and timescales and reduce the concentration of terrigenous nutrients, which will directly change the primary productivity of shelf seas. Meanwhile, the absence of tides significantly increased the seawater salinity of the shelf seas. The total alkalinity of seawater is controlled mainly by salinity (Millero et al., 1998), dominating the air-sea $CO_2$ flux (Kantha, 2004). Thus, the change in primary productivity and the bias in seawater salinity could ultimately lead to errors in the simulation of the $CO_2$ flux and carbon cycle in shelf seas in climate models. Third, the terrigenous organic and inorganic carbon carried by rivers is over 0.8 PgC/year (1 Pg = $10^{15}$ g) and is input into shelf seas (Bauer et al., 2013). The simulation of the export rate of river water and terrigenous carbon in shelf seas in a no-tide model would be falsely accelerated, which may reduce terrigenous carbon burial and emission in the shelf sea and thus increase the amount of terrigenous carbon entering the open ocean (Schlünz & Schneider, 2000). In these ways, the absence of tides in climate models could induce a strong uncertainty in the simulation of the carbon cycle and thus affect the accuracy of the climate projection of the climate models and the earth system models. However, the effect of the change of the river water behavior in no-tide models on the bias of the carbon cycle simulation and climate projection still need to be further quantitatively studied using a high-resolution earth system model.

## 5 Conclusions

Using numerical modeling and sensitivity experiments, this study assessed the effect of tides on river water transport pathways and timescales and the water concentration of three major rivers (i.e., the Yellow, Yalujiang, and Changjiang Rivers) in the ESSC. The model results suggest that tides induced more dispersed transport pathways for the Yellow and Yalujiang Rivers, but more concentrated transport pathways for the Changjiang River. By weakening the shelf currents, tides increased the water transport timescales of the three rivers by 2–10 times and thus significantly slowed the transport and export of the river water over the shelf. The slow export of river water induced by tides increased the river water concentration by approximately one order in magnitude and decreased seawater salinity in the ESSC. Moreover, the effects of tides on river water behavior were stronger in relatively enclosed seas (i.e., the Bohai and Yellow Seas) than in relatively open seas (i.e., the East China Sea). Given the important role of river water in water and carbon cycling, climate and earth

system models without tides may be biased in simulating and predicting the global water and carbon cycle. Therefore, the effects of tides on river water behaivor should be carefully considered in climate and earth system modeling.

## Code availability

The source code of numerical model used in this study is available on request. Please contact Lei Lin (llin@sdust.edu.cn).

## Data availability

The model data are shown in the figures in the main text which is available on request. Please contact Lei Lin (llin@sdust.edu.cn).

## Author contributions

LL, HL, and QF conducted the numerical experiments and analyzed the data. LL, HL, and XH wrote the initial draft, and all authors have contributed to editing the paper. XG provided the hydrodynamic model. All authors contributed to discussing and interpreting the results.

## Competing interests

The contact author has declared that neither they nor their co-authors have any competing interests.

## Acknowledgments

The authors thank Prof. Hui Wu and the two anonymous reviewers for their very constructive comments that helped us improve the manuscript. This study was supported by the National Key Research and Development Program of China(2020YFA0607900) and the National Natural Science Foundation of China (Grant No. 42125503 and 41706011). Xinyu Guo was supported by Grants-in-Aid for Scientific Research (MEXT KAKENHI grant numbers: 20H04319).

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
