# Peer review of "Effect of tides on river water behavior over the eastern shelf seas of China"

_Hydrology and Earth System Sciences, 2022_

## Author Comment (AC1)

**Reply to Reviewer #1**

Dear Reviewer,

We thank you very much for your very insightful and constructive comments to improve our manuscript. According to your comments, we have carefully revised the manuscript. Please see the detailed responses to each comment below. The comments are cited in black. The response to each comment is set out in blue. The revised manuscript will be submitted at the stage of revision.

Lei Lin

Using transport pathway, time scale and the concentration as indicators, this study examined the influence of tides on the river water behaviours over the shelf of the marginal sea. The results are important to understand and improve the climate model in the global water and biogeochemical cycles. The methods are solid, and the results are reasonable. Several points can be further clarified at this stage.

1. Several indicators including the pathway, time scales, tracer concentrations were used in this study. (1) The links and discrepancies among those indicators can be further explained, thus give better illustration on the influence of tide. (2) Such as there is high water age in the region of YSCWM, which trapped the river waters, while the emergence possibility there is relatively low.
   **Response:** (1) Following your instruction, we have added an extra paragraph to explain the physical meanings of the different indicators and their links and discrepancies. The emerging probability of the particles reflects the proportion of river water that passes through the region, which is mainly related to the shelf currents. The regions with higher emerging probability indicate a greater amount of river water passing through them and thus are the main pathway of river water transport. The transit area can be an approximate representation of the area through which the river flows. Water age reflects how long the river water has been on the shelf after leaving the estuary, while the transit time reflects the total time the river

water takes from entering to leaving the shelf. The tracer concentration reflects the stock of the river water over the shelf, which can influence the salinity field. It should be noted that the magnitudes of the emerging probability, water age, and tracer concentration may not be directly related. For instance, a small proportion of one river water may stay in region A for a long time while the most proportion of the river water may quickly leave the shelf through region B. Then, region A would obtain a relatively large water age but a relatively small emerging probability compared with region B. However, the tracer concentration in region A could be either high or low which is determined by the accumulated amount of river water in this region.

(2) The high age of river water in the central Yellow Sea actually reflects the slow water exchange of the Yellow Sea. Once a particle goes into this region, it needs a long time to leave. However, only a small proportion of the river water stayed in the region of the YSCWM, and most of the river waters were transported along the west coast of the Korean peninsula. Meanwhile, the area of the central Yellow Sea is large. Thus, in spite of a high water age, the emerging probability at the region of the YSCWM was low. We have added this comment to the revised manuscript.

2. The transit time shows high STD, which is comparable to the mean values. There is also remarkable seasonal variability. While those variabilities were not well discussed or explained.

   **Response:** The high STD could be related to the water mixing and the high spatial variation in the water residence time. As shown by Lin et al. (2022), there is significant spatial variation in the water residence time (WRT) on the eastern shelf of China. The WRT can reach 15 years in the Bohai Sea, 4-10 years in the Yellow Sea, and about 1 year in the East China Sea. The particles released at the estuaries were continuously dispersed as they move under the effect of mixing. Dispersed particles can go into different seas and get different residence times over the shelf, which can explain the high STD of the transit time. The seasonal variability of the transit time could be related to the seasonal variation in the shelf currents. For instance, the part of the Changjiang River water in spring is prone to be transported into the Yellow Sea under the effect of the southerly wind and thus obtained a

longer retention time over the shelf than in other seasons. Meanwhile, the large range of the residence time of the Yellow Sea induced the large STD in spring. We have added this comment to the revised manuscript.

3. How the emergence possibility represents the "pathway" of the river water? The emergence possibility is discontinuous spatially, should give a better explanation.

**Response:** The discontinuity of the emerging probability of the river particles could be due to the spatial variation in the area through which the river water flows, which can be understood using a schematic diagram as below (Figure S1). Because the total number of particles is constant, the emerging probability of the particles should be relatively large when they go through a small-area region, and vice versa. The area is relatively small in the regions of the inlet and outlet and is relatively large in the middle of the shelf sea. Thus, the emerging probability seems to be discontinued at the middle of the river water transport pathway, e.g., in the case of the Changjiang River. We have added this comment to the revised manuscript.

[Figure]

**Figure S1.** Schematic diagram for the spatial variation in the emerging probability. The area in the middle shelf is relatively large, resulting in a relatively low emerging probability at grid cells than in the regions of the inlet and outlet.

4. Why the tidal effect induced more dispersed transport for the Yellow and Yalujiang Rivers' waters but more concentrated transport for the Changjiang River water?

**Response:** The different tidal effects could be related to the different water transport patterns of the rivers. For the Yellow and Yalujiang Rivers water, the west coast of the Korea Peninsula with strong tides and cross-shelf tidal currents (Figure S2) is their main transport pathway (Figure 2 in the manuscript). As shown in Figure S3, without considering tides, the southward coastal current along the west

coast of the Korean Peninsula was intensified, especially in winter, which accelerated the river water export and reduced the transport timescales of the Yellow and Yalujiang Rivers' waters. When considering the tides, on one hand, the cross-shelf tidal currents along the coast can increase the cross-shelf water dispersion as the magnitude of tidal dispersion is proportional to the tidal velocity (Geyer & Signell, 1992). On the other hand, the weakened coast current slowed down the along-shore transport of the river waters and further promoted the cross-shelf water dispersion. Thus, compared to the NoTide case, the Control run obtained a more dispersed transport for the Yellow and Yalujiang Rivers' waters. For the Changjiang River, there are three major branches for the river water transport, i.e., the northeastward branch to Cheju Island, the northward branches to the Jiangsu coast, and the southward branch to the Zhejiang coast (Wu et al., 2014). The northeastward branch to Cheju Island is the dominant one for the Changjiang River water transport. In the NoTide case, the intensified coastal currents could increase the river water transport to the Yellow Sea along the Jiangsu coast in summer and to the East China Sea along the Zhejiang coast in winter, which intensified the northward and southward branches of the Changjiang River water transport. Thus, the water particles of the Changjiang River in the NoTide case were more dispersed than those in the Control run. We have added this discussion to the revised manuscript.

[Figure]

Fig. S2. (a) Tidal ellipses for $M_2$ tide. (b) Contour lines of coamplitude (solid lines, with an interval of 0.2 m) and cophase (dashed lines, with an interval of 30°) for $M_2$ tide.

[Figure]

Figure S3. The changes in shelf currents in February (a) and August (b) after removing tides in the model.

[Figure]

Figure S4. Schematic map of the shelf currents and the major pattern of the river water transport for the Control run (a) and NoTide case (b). The blue and red arrows denote the coastal currents and warm currents, respectively. The blue dashed arrow at the Jiangsu Coast denotes the coastal current in summer. The thickness of the arrows denotes the intensity of the shelf currents. The black dashed wavy line in (a) denotes the cross-shelf dispersion induced by tides. The black arrows denote the direction and the branches of the river water transport.

5.  The tide has different effect on the river behaviour in Bohai& Yellow Seas and the East China Sea, due to the semi-enclosed geometry or the water depth?

    **Response:** The different effects of tides on the river behaviour in the Bohai and Yellow Seas and the East China Sea could be related to both the geometry and the water depth, as both of them can influence the water exchange and the strength of the tidal current. The semi-enclosed geometry in the Bohai and Yellow Seas induced relatively weak water exchange compared to the relatively open East China Sea. Thus, the coastal currents are important dynamics for the water exchange in the Bohai and Yellow Seas and the tidal effect on the coastal currents significantly influenced the river water behaviour in the Bohai and Yellow Seas.

On the other hand, the change in the bottom friction induced by the tides is an important mechanism for the tidal effect on the shelf currents. Due to the larger inertia and stronger stratification, the shelf currents in relatively deep water could be less influenced by changes in bottom friction than that in relatively shallow water (Figure S5). Thus, the effect of tides on river water behavior in the East China Sea was relatively smaller than those in the Yellow and Bohai seas. We have added this comment to the revised manuscript.

[Figure]

Figure S5. The relative changes in the annual mean magnitude of shelf currents after removing the tides in the model. Positive (negative) values indicate an increase (decrease) in the magnitude of the currents.

Some minor comments:

1.  What's the frequency of the hydrodynamic model data used to drive the particle tracking and passive tracer model.

**Response:** The hydrodynamic model data with a time interval of 0.5 hours were used to drive the particle tracking and passive tracer models. We have added this information to the revised manuscript.

2. For the transit time and water age, did authors consider the re-entry processes?

**Response:** The re-entry process was not considered in the calculation of the transit time and water age. Thus, the timescales in this study represent the time from leaving the estuaries to touching the shelf boundary for the first time. We have added this information to the revised manuscript.

3. The changes of mixing intensity can be shown

**Response:** We have added a figure (Figure S6) to show the changes in the vertical turbulent diffusivity coefficients in the revised manuscript. Due to the space limitation, we put this figure in the supplementary material.

[Figure]

Figure S6. The relative change in the vertical turbulent diffusivity coefficients in February (a) and August (b) after removing tides in the model.

4. Line 165: "The Yalujiang River water passes mainly through the western Yellow Sea", western or eastern?

**Response:** It should be "eastern". We have revised this word in the revised manuscript.

5. Figure 10 should be improved to make the arrows more clear

**Response:** We have redrawn this figure (see Figure S7 below) in the revised manuscript to make the arrows clearer.

[Figure]

Figure S7. Monthly and vertically averaged velocity for the Control run (left panels) and No-Tide case (right panels), respectively. (a, c) for February and (b, d) for August.

6. In some figures, the STD is too larger. Refine the figure to make it clear

**Response:** We have redrawn these figures (see Figure S8 below) in the revised manuscript to make it clear.

[Figure]

Figure S8. The daily mean water transit time over the shelf of the three rivers for the Control run (blue) and PMT-Tide case (red). STD in the figure label denotes the standard deviation of the transit time for particles released on the same day.

Reference:

Geyer, W. R., & Signell, R. P. (1992). A reassessment of the role of tidal dispersion in estuaries and bays. Estuaries, 15(2), 97-108.

Lin, L., Liu, D., Guo, X., Luo, C., Cheng, Y. (2020). Tidal Effect on Water Export Rate in the Eastern Shelf Seas of China. Journal of Geophysical Research: Oceans, 125(5).

Wu, H., Zhu, J., Shen, J., Wang, H. (2011). Tidal modulation on the Changjiang River plume in summer. Journal of Geophysical Research: Oceans, 116(C8).

Wu, H., Shen, J., Zhu, J., Zhang, J., Li, L. (2014). Characteristics of the Changjiang plume and its extension along the Jiangsu Coast. Continental Shelf Research, 76, 108-123.

---

## Author Comment (AC2)

Dear Prof. Hui Wu,

We thank you very much for your very insightful and constructive comments to improve our manuscript. According to your comments, we have carefully revised the manuscript. Please see the detailed responses to each comment below. The comments are cited in black. The response to each comment is set out in blue. The revised manuscript will be submitted at the stage of revision.

Lei Lin

In this paper the authors investigated the tidal effects on freshwater transport in shelf seas of China with a numerical model. The results indicated that tide slows down the shelf currents and increases the time scale of freshwater retained in the shelf seas. Then the authors argued that only parameterizing the tidal mixing in model cannot faithfully reflect the actual tidal effects, thus they suggest the climate models should include the tidal forcing in an explicit way. Over all the scientific point of this paper is very clear and the conclusion is well supported by the model results. I enjoyed in reading this paper and recommend to accept it after some revisions.

1.  My major concern is on the dynamics. The authors attribute the slowdown of shelf current under tidal effects to the enhanced bottom stress. This mechanism was discussed too in their earlier paper (Lin et al., 2020, JGR) by citing the results of previous papers. I should say that this mechanism applies more or less to barotropic region only. In stratified region, tidal can also adjust the baroclinic structure thus affect the currents in other ways. It is a complicated process and the authors cannot simply attribute it to the enhanced effective friction. More discussions are necessary besides the sensitivity experiments, and also the authors should cite the seminal paper instead of just Lin et al. (2020).

    **Response:** Following your comment, we discussed the effect of tides on the shelf currents in-depth and cited more seminal papers to support the analysis in the

revised manuscript.

The comparison of the shelf currents between the Control run and NoTide case shows that the change in the shelf currents mainly occurred in the coastal region and the central Yellow Sea (Figures S1a-S1d), and the change in winter is more significant than that in summer (Figures S1e-S1f). The analysis of water transport fluxes by Lin et al. (2020) showed the winter season accounted for ~73% of the volume transport for the entire year and suggested that the winter processes dominate the water exchange of the Yellow Sea. Due to the strong surface cooling, the coastal water (even in the central Yellow Sea) in winter was well-mixed. Thus, the change in the barotropic process could dominate the change of the shelf currents. In addition, the change in the coastal current along the Jiangsu and Zhejiang coasts shows southward in winter and northward in summer (Figure S1e and S1f), which are consistent with the directions of the seasonal wind-driven currents (northerly wind in winter and southerly wind in summer). This indicates that the wind-driven coastal current was significantly weakened by tides, which can be explained by the nonlinear interaction between tides and wind-driven current in a quadratic bottom friction term (e.g., Moon, 2005; Wu et al., 2018). When considering tides, the period-mean bottom resistance becomes about one order larger due to the quadratic nature of the bottom friction and thus greatly reduced the wind-driven coastal currents (Wu et al., 2018). Besides the tidally enhanced bottom resistance, the tidally enhanced mixing and induced residual current and residual transport could also modulate the shelf currents, especially during the stratified season (e.g., Moon, 2005; Lie and Cho, 2016; Moon et al., 2009; Wu et al., 2011; Wu et al., 2014; Wu and Wu, 2018). Moon et al (2009) suggested that the tidal forcing induces a strong southward residual flow along the western slope of the Yellow Sea in summer, which can explain the change of northward current in the NoTide case in the middle of the Yellow Sea (Figure S1f). Wu et al. (2011) suggested that tidally enhanced mixing resulted in a strong horizontal salinity gradient at the north of the Changjiang River mouth, which acted as a dynamic barrier and restricts the northward transport of the Changjiang River water to the Jiangsu coast in summer.

In addition, strong tidal mixing induces a mixing front along the Zhe-Min coast and maintains a down-shelf frontal current (Wu and Wu, 2018), which is important for the southward transport of the Changjiang River water. Overall, the tidal effect on the shelf current is complicated and diverse. However, the tidally enhanced bottom stress should be the dominant mechanism due to the important role of winter currents in the water exchange and the significant change in the winter coastal current in the Control run and NoTide case.

[Figure]

Figure S1. Comparison of the shelf currents between the Control run and NoTide case. (a) and (b) are the annual mean and vertically averaged velocities for the Control run and No-Tide case, respectively. The arrow and color in (a) and (b) denote the direction and magnitude of the velocity, respectively. (c) and (d) show

the change and relative change in the magnitude of the shelf currents after removing tides in the model. (e) and (f) show the changes in shelf currents in February and August, respectively.

2. Between Line 285-290 the authors stated that "the decrease in coastal currents results in a smaller gradient of sea surface height across the continental shelf, thus weakening Yellow Sea Warm Current in the control run compared to the no-tide run". This mechanism needs justification because (1) under the tidal effects the currents is no longer geostrophic (2) it is unclear why the decreased SSH gradient would weaken the YSWC, mechanism and citations are needed. I understand such a mechanism could be discussed in previous results, but here you might give a clear explanation.

**Response:** Following your comment, we explained the effect of tides on the Yellow Sea Warm Current in-depth and cited more papers to support the discussion in the revised manuscript.

(1) Yes, under the tidal effects the high-frequency currents were no longer geostrophic. However, the Yellow Sea Warm Current (YSWC) in this study was discussed on the timescale of as least one month (i.e., only the subtidal transport is concerned), and the YSWC can be considered quasi-geostrophic on the timescale of months. For instance, the model results in Moon (2005) showed that the momentum at the YSWC in February was balanced by the pressure gradient, Coriolis force, and vertical vorticity term when considering tides (Figure 11 in Moon, 2005). In addition, tides might influence subtidal transport by inducing tidal stress. Wu et al. (2018) showed that the tidal stress mainly occurred in the shallow coastal water and less influenced the central Yellow Sea (Figure 10 in Wu et al. 2018). Thus, the YSWC on the subtidal timescale should be quasi-geostrophic even under the tidal effect.

(2) Several studies have interpreted the formation of the YSWC using the upwind theory (e.g., Hsueh and Yuan, 1997; Isobe, 2008; Lin et al., 2011). The mechanism is outlined below. In the semi-enclosed Yellow Sea, the wind stress is the dominant

forcing in winter and thus the currents along the eastern and western coasts are in the same direction as the northerly wind. The outward coastal currents on both sides can reduce the water elevation inside, pile up water mass at the downwind end, and form the negative pressure gradient (decrease to the north). Along the trough of the central Yellow Sea, the water is deep. The depth-averaged wind stress is small and too weak to counter the northward pressure gradient force. This imbalance results in a pressure-driven flow that is opposite to the direction of the surface wind along the trough of the Yellow Sea. Under the effect of the Coriolis force, the upwind current is further geostrophically adjusted and slightly shifted to the western side of the trough to satisfy the balance of the potential vorticity (Lin et al., 2011). When considering tides, enhanced bottom friction weakened the coastal currents and thus decreased the water elevation difference and thus the pressure gradient (Moon 2005; Lin et al., 2020). Meanwhile, the enhanced bottom friction below the YSWC further counters a part of the northward momentum (Moon 2005). Thus, the weaker YSWC was presented in the model with tides. The response of the YSWC on the change in bottom friction was also reproduced in an idealized semi-enclosed region and more detail is discussed by Lin et al. (2011).

3. Line 190 "suggesting the YSCWM region could trap river water for several years". YSCWM is in the bottom, how could it trap the surface river water? Mechanism should be given.

**Response:** Yes, the YSCWM occurred at the bottom of the Yellow Sea during the stratified season. It formed mainly due to the retention of the winter cold water (Zhang et al., 2008). The water in winter is well mixed in the Yellow Sea (e.g., Zhu et al., 2018; Lin et al., 2019). Thus, some surface river water in the Yellow Sea could be mixed in the whole water column in winter and partially stay at the bottom during the stratified season. We have added this comment to the revised manuscript.

4. Can you provide a more in-depth explanation as to why tides induced more dispersed transport pathways for the Yellow and Yalujiang Rivers, but more concentrated transport pathways for the Changjiang River? It is very interesting to know the underlying mechanism.

**Response:** The mechanism is summarized in Figure S3. The different tidal effects should be related to the different water transport patterns of the rivers. For the Yellow and Yalujiang Rivers water, the west coast of the Korea Peninsula with strong tides and cross-shelf tidal currents (Figure S2) is their main transport pathway (Figure 2 in the manuscript). As shown in Figure S1e, without considering tides, the southward coastal current along the west coast of the Korean Peninsula was intensified, especially in winter, which accelerated the river water export and reduced the transport timescales of the Yellow and Yalujiang Rivers' waters. When considering the tides, on one hand, the cross-shelf tidal currents along the coast can increase the cross-shelf water dispersion as the magnitude of tidal dispersion is proportional to the tidal velocity (Geyer & Signell, 1992). On the other hand, the weakened coast current slowed down the along-shore transport of the river waters and further promoted the cross-shelf water dispersion. Thus, compared to the NoTide case, the Control run obtained a more dispersed transport for the Yellow and Yalujiang Rivers' waters. For the Changjiang River, there are three major branches for the river water transport, i.e., the northeastward branch to Cheju Island, the northward branches to the Jiangsu coast, and the southward branch to the Zhejiang coast (Wu et al., 2014). The northeastward branch to Cheju Island is the dominant one for the Changjiang River water transport. In the NoTide case, the intensified coastal currents could increase the river water transport to the Yellow Sea along the Jiangsu coast in summer and to the East China Sea along the Zhejiang coast in winter, which intensified the northward and southward branches of the Changjiang River water transport. Thus, the water particles of the Changjiang River in the NoTide case were more dispersed than those in the Control run. We have added this discussion to the revised manuscript.

[Figure]

Fig. S2. (a) Tidal ellipses for M2 tide. (b) Contour lines of coamplitude (solid lines, with an interval of 0.2 m) and cophase (dashed lines, with an interval of 30°) for M2 tide.

[Figure]

Figure S3. Schematic map of the shelf currents and the major pattern of the river water transport for the Control run (a) and NoTide case (b). The blue and red arrows denote the coastal currents and warm currents, respectively. The blue dashed arrow

at the Jiangsu Coast denotes the coastal current in summer. The thickness of the arrows denotes the intensity of the shelf currents. The black dashed wavy line in (a) denotes the cross-shelf dispersion induced by tides. The black arrows denote the direction and the branches of the river water transport.

**Reference:**

Geyer, W. R., & Signell, R. P. (1992). A reassessment of the role of tidal dispersion in estuaries and bays. Estuaries, 15(2), 97-108.

Hsueh, Y., & Yuan, D. (1997). A numerical study of currents, heat advection, and sea-level fluctuations in the Yellow Sea in winter 1986. Journal of physical oceanography, 27(11), 2313-2326.

Isobe, A. (2008). Recent advances in ocean-circulation research on the Yellow Sea and East China Sea shelves. Journal of oceanography, 64(4), 569-584.

Lie, H. J., & Cho, C. H. (2016). Seasonal circulation patterns of the Yellow and East China Seas derived from satellite-tracked drifter trajectories and hydrographic observations. Progress in Oceanography, 146, 121-141.

Lin, L., Liu, D., Guo, X., Luo, C., Cheng, Y. (2020). Tidal Effect on Water Export Rate in the Eastern Shelf Seas of China. Journal of Geophysical Research: Oceans, 125(5).

Lin, L., Wang, Y., Liu, D. (2019). Vertical average irradiance shapes the spatial pattern of winter chlorophyll-a in the Yellow Sea. Estuarine, Coastal and Shelf Science, 224, 11-19.

Lin, X., Yang, J. (2011). An asymmetric upwind flow, Yellow Sea Warm Current: 2. Arrested topographic waves in response to the northwesterly wind. Journal of Geophysical Research Atmospheres, 116(C4), 5.

Moon, I. (2005). Impact of a coupled ocean wave–tide–circulation system on coastal modeling. Ocean Modelling, 8(3), 203-236.

Moon, J. H., Hirose, N., & Yoon, J. H. (2009). Comparison of wind and tidal contributions to seasonal circulation of the Yellow Sea. Journal of Geophysical

Research: Oceans, 114(C8).

Wu, H., Gu, J., Zhu, P. (2018). Winter Counter - Wind Transport in the Inner Southwestern Yellow Sea. Journal of Geophysical Research: Oceans, 123(1), 411-436.

Wu, H., Shen, J., Zhu, J., Zhang, J., Li, L. (2014). Characteristics of the Changjiang plume and its extension along the Jiangsu Coast. Continental Shelf Research, 76, 108-123.

Wu, T., Wu, H. (2018). Tidal Mixing Sustains a Bottom-Trapped River Plume and Buoyant Coastal Current on an Energetic Continental Shelf. Journal of Geophysical Research: Oceans, 123(11), 8026-8051.

Zhang, S. W., Wang, Q. Y., Lü, Y., Cui, H., Yuan, Y. L. (2008). Observation of the seasonal evolution of the Yellow Sea Cold Water Mass in 1996–1998. Continental Shelf Research, 28(3), 442-457.

Zhu, J., Shi, J., Guo, X., Gao, H., & Yao, X. (2018). Air-sea heat flux control on the Yellow Sea Cold Water Mass intensity and implications for its prediction. Continental Shelf Research, 152, 14-26.

---

## Author Comment (AC3)

Dear Reviewer,

We thank you very much for your very insightful and constructive comments to improve our manuscript. According to your comments, we have carefully revised the manuscript. Please see the detailed responses to each comment below. The comments are cited in black. The response to each comment is set out in blue. The revised manuscript will be submitted at the stage of revision.

Lei Lin

In this paper, the authors use POM with an offline particle-tracking model to reveal the behaviors of the fluvial fresh water discharged from three major rivers in the eastern shelf seas of China. This is an important study that may serve to better understand influences of tides on transit time, water ages, and pathways of the fluvial fresh water on the shelf seas, and could further help in calibration/validation of the tidal parameterizations for the Earth system models. The results of the particle-tracking model clearly demonstrate that the tides change the behaviors of river water. But I struggled to interpret some of these behaviors shown with different types of tracers. With additional clarifications in the hydrodynamic processes on which the spatial distributions of tracers rely, and more details in model setups and methodology, the readability of this paper could be largely improved. For this reason, I would recommend returning this manuscript to the authors for revision.

**Response:** Following your instruction, we have added an extra paragraph to explain oceanographic processes or properties that different types of tracers represent and more details on model setups and methodology to increase the readability of this paper.

The following are detailed comments:

1. Line 77: "… from the perspectives of transport pathways, transport timescales, and water concentration distribution". This is a general comment about these river water behaviors. The authors represent details in each behavior with tracer results, but do not explain well what oceanographic processes or properties they are presented. For example, as RC1 commented, why the water ages are high in the Yellow Sea, but the emerging probabilities are low? It may help to have an extra paragraph to explain oceanographic processes or properties that different types of

tracer represent, such as the water age reflects the residence time, emerging probability depends probably on both buoyancy and atmospheric forcings, and the tracer concentrations could show the effects in salinity field, etc. Then the effects of tides can be rooted through differences in tracer fields for physical reasons.

**Response:** Following your instruction, we have added an extra paragraph to explain oceanographic processes or properties that different types of tracers represent. The emerging probability of the particles reflects the proportion of river water that passes through the region, which is mainly related to the shelf currents. The regions with higher emerging probability indicate a greater amount of river water passing through them and thus are the main pathway of river water transport. The transit area can be an approximate representation of the area through which the river flows. Water age reflects how long the river water has been on the shelf after leaving the estuary, while the transit time reflects the total time the river water takes from entering to leaving the shelf. The tracer concentration reflects the stock of the river water over the shelf, which can influence the salinity field. It should be noted that the magnitudes of the emerging probability, water age, and tracer concentration may not be directly related. For instance, a small proportion of one river water may stay in region A for a long time but the most proportion of the river water may quickly leave the shelf through region B. Then, region A would obtain a relatively large water age but a relatively small emerging probability compared with region B. However, the tracer concentration in region A could be either high or low which is determined by the accumulated amount of river water in this region.

2. Line 91: In general, it would improve the readability with some level of detail in model setups. The authors provide a list of references containing very detailed information about the model they used, but I must stop reading and search for them.

**Response:** We have added more details to model setups in the revised manuscript, including the tidal constituents, the vertical layers, the external forcings, etc.

3. Line 94: What are these tidal constituents?

**Response:** The major four tidal constituents considered in the model were $M_2$, $S_2$, $K_1$, and $O_1$. We have added this information to the revised manuscript.

4. Line 95: How many vertical layers does POM have?

**Response:** The model had 21 non-uniform sigma layers in the vertical direction with finer resolution in the upper layer. We have added this information to the revised manuscript.

5. Line 96: What forcings are used in the POM; are wind, radiation, and evaporation considered?

**Response:** The forcings used in the model included winds, heat fluxes, precipitation, and evaporation. We have added this information to the revised manuscript.

6. Line 125-126: "… by dividing the number of particles emerging in the grid cell by the total number of particles released." Is the denominator the total tracer released per day, or for the entire simulation of 30 years?

**Response:** The denominator is the total tracer released per day, i.e., one thousand. We have added this information to the revised manuscript.

7. Line 148: Does reenter account on the shelf boundary, or is t1 accounted for the first leave?

**Response:** The $t_1$ was accounted for the first leave. The re-entry process was not considered in both the transit time and water age. We have added this information to the revised manuscript.

8. Line 152-153: (1) This should be mentioned earlier in 2.1, and emphasize the tracer model is decoupled from the hydrodynamic model, as I thought the tracer module was running with POM. (2) Are three years long enough for POM to get equilibrium of the water exchanges along the shelf boundary? (3) If the hydrodynamic fields are not in a steady state, would it affect the results of particle-tracking model? (4) Which outputs of POM are used to force the particle-tracking model? (5) Does POM consider evaporation? (6) Does tracer-tracking model consider surface sink?

**Response:** (1) Following your suggestion, we have added this information in Section 2.1. (2) Yes, it was long enough for the model to get equilibrium of water exchanges as the initial field used in the model was derived from the model that has reached equilibrium provided by Wang et al. (2008). The results of the total

kinetic energy of the model are shown in Figure S1, demonstrating that the model quickly entered steady seasonal cycles. (3) Yes, it will affect the results to some extent. (4) The sea level, water depth, velocity, and diffusivity coefficients outputted from POM were used to force the particle-tracking and tracer models. (5) Yes, POM has considered evaporation. (6) But the surface sink was not considered in the particle-tracking and tracer models. We have added these revisions and information in the revised manuscript.

[Figure]

Figure S1. (a) The 3-year variation in the total kinetic energy of the models for the Control run and NoTide case, respectively. (b) The variation in the total kinetic energy in the first month.

9. Line 170: "Figure 2 …" (1) Is the emerging probability calculated only for surface layer, or integrated vertically? (2) Why is the spatial pattern of emerging probability discontinued at Yellow Sea trough for Changjiang River?

    **Response:** (1) The emerging probability calculated for integrated vertically. We have added this information in Section Methods of the revised manuscript.

    (2) The discontinuity of the emerging probability of the Changjiang River particles could be due to the spatial variation in the area through which the river water flows, which can be understood using a diagram below (Figure S2). Because the total number of particles is constant, the emerging probability of the particles should be relatively large when they go through a small-area region, and vice versa. The area is relatively small in the regions of the inlet and outlet and is relatively large in the

middle of the shelf sea. Thus, the emerging probability seems to be discontinued at the central Yellow Sea. The dynamic mechanism of the discontinuity of the emerging probability could be related to the Yellow Sea Warm Current (YSWC) along the Yellow Sea trough. The northward YSWC brings some Changjiang River water into the Yellow Sea and induced a more dispersed transport over the shelf, thus the rapid reduction probability at the Yellow Sea trough. We have added this comment to the revised manuscript.

[Figure]

**Figure S2.** Schematic diagram for the spatial variation in the emerging probability. The area in the middle shelf is relatively large, resulting in a relatively low emerging probability at grid cells than in the regions of the inlet and outlet.

10. Line 199: "Figure 3 …" It would be helpful to (1) show the annual mean vertically averaged velocities, and to (2) explain why the water ages are high in Yellow Sea, but the emerging probability is low.

    **Response:** (1) We have added a figure (Figure S3) to the revised manuscript to show the annual mean vertically averaged velocities of the Control run and NoTide case.

[Figure]

Figure S3. The annual mean and vertically averaged velocities for the Control run (a) and No-Tide case (b). The arrow and color denote the direction and magnitude of the velocity, respectively.

(2) The high age of river water in the central Yellow Sea actually reflects the slow water exchange of the Yellow Sea. Once a particle goes into this region, it needs a long time to leave. However, only a small proportion of the river water stayed in the region of the YSCWM, and most of the river waters were transported along the west coast of the Korean peninsula. Meanwhile, the area of the central Yellow Sea is large. Thus, in spite of a high water age, the emerging probability at the region of the YSCWM was low. We have added this comment to the revised manuscript.

11. Line 213: Consider replacing Figure 4 with Figure 9, as both contain the transit time for "control" and "no-tide" cases.

    **Response:** Thanks for your suggestion. For the convenience of comparison, the means of the transit time for the Control run and NoTide case were plotted in Figure 9. However, their standard deviations were not included due to the space limitation of the figure. Thus, we consider that Figure 9 cannot completely replace Figure 4.

12. Line 225: "Figure 5 …" Why the tracer concentrations are low in Cheju Strait for all rivers, but their emerging probabilities are high?

    **Response:** Since the Cheju Strait is an important outlet for river water, most of the river particles would pass through the strait which induced the high emerging

probabilities of the river water at the Cheju Strait. However, when the river water was transported to the Cheju Strait, it has been mixed with shelf seawater which decreased the river water concentration. Thus, we obtained high emerging probabilities but low tracer concentrations in Cheju Strait. We have added this comment to the revised manuscript.

13. Line 297: "Figure 10 …" (1) Please consider changing the color scheme to increase readability. (2) As all tracer fields are shown in annual mean, what are the net effects of these seasonal differences in shelf currents? Generally, a set of annual mean vertically integrated plots of velocities and/or tidal current ellipses would greatly help better understand the tracer patterns.

**Response:** (1) Following your instruction, we have changed the color scheme of Figure 10 (Figure S4 below) to increase readability.

(2) We also added figures (Figures S3 and S5) for the annual mean vertically integrated velocities and the tidal current ellipse ($M_2$ tidal constituent) to help understand the net effects of the seasonal differences in shelf currents on the tracer patterns. The change in vertically integrated velocity for the annual mean (Figure S3) is very close to that for the winter (Figure S4a and S4b), suggesting that the winter processes dominated the tracer pattern as the stronger shelf currents and water exchange occurred in winter.

[Figure]

Figure S4. Monthly and vertically averaged velocity for the Control run (left panels) and No-Tide case (right panels), respectively. (a, c) for February and (b, d) for August.

[Figure]

Fig. S5. (a) Tidal ellipses for M2 tide. (b) Contour lines of coamplitude (solid lines, with an interval of 0.2 m) and cophase (dashed lines, with an interval of 30°) for M2 tide.

14. Line 309-310: The seasonal velocities (Figure 10) also show differences in mesoscale eddies. Would they affect the river water behaviors?

Response: Insightful question. The flow field at the middle shelf in the Control run showed a pattern of a mesoscale eddy which might trap some river water and reduce the dispersion of the water. We have added this comment to the revised manuscript.